# Leveraging inter-individual transcriptional correlation structure to infer discrete signaling mechanisms across metabolic tissues

Mingqi Zhou[1,2†], Ian Tamburini[1,2†], Cassandra Van[2†], Jeffrey Molendijk[3], Christy M Nguyen[1,2], Ivan Yao-Yi Chang[1], Casey Johnson[1,2], Leandro M Velez[1,2], Youngseo Cheon[1,2], Reichelle Yeo[4], Hosung Bae[1,2], Johnny Le[1,2], Natalie Larson[1,2], Ron Pulido[1,2], Carlos HV Nascimento-Filho[1,2], Cholsoon Jang[1,2], Ivan Marazzi[1,2], Jamie Justice[5], Nicholas Pannunzio[6], Andrea L Hevener[7,8], Lauren Sparks[4], Erin E Kershaw[9], Dequina Nicholas[1,2,10], Benjamin L Parker[3], Selma Masri[1,2], Marcus M Seldin[1,2]*

[1]Department of Biological Chemistry, UC Irvine, Irvine, United States; [2]Center for Epigenetics and Metabolism, UC Irvine, Irvine, United States; [3]Department of Anatomy and Physiology, University of Melbourne, Melbourne, Australia; [4]Translational Research Institute, AdventHealth, Orlando, United States; [5]Veterans Administration Greater Los Angeles Healthcare System, Geriatric Research Education and Clinical Center (GRECC), Los Angeles, United States; [6]Divison of Hematology/Oncology, Department of Medicine, UC Irvine Health, Irvine, United States; [7]Department of Medicine, Division of Endocrinology, Diabetes, and Hypertension, David Geffen School of Medicine at UCLA, Los Angeles, United States; [8]Iris Cantor-UCLA Women's Health Research Center, David Geffen School of Medicine at UCLA, Los Angeles, United States; [9]Division of Endocrinology, Department of Medicine, University of Pittsburg, Pittsburgh, United States; [10]Department of Molecular Biology and Biochemistry, School of Biological Sciences, University of California Irvine, Irvine, United States

*For correspondence:
mseldin@uci.edu

†These authors contributed equally to this work

**Abstract** Inter-organ communication is a vital process to maintain physiologic homeostasis, and its dysregulation contributes to many human diseases. Given that circulating bioactive factors are stable in serum, occur naturally, and are easily assayed from blood, they present obvious focal molecules for therapeutic intervention and biomarker development. Recently, studies have shown that secreted proteins mediating inter-tissue signaling could be identified by 'brute force' surveys of all genes within RNA-sequencing measures across tissues within a population. Expanding on this intuition, we reasoned that parallel strategies could be used to understand how individual genes mediate signaling across metabolic tissues through correlative analyses of gene variation between individuals. Thus, comparison of quantitative levels of gene expression relationships between organs in a population could aid in understanding cross-organ signaling. Here, we surveyed gene-gene correlation structure across 18 metabolic tissues in 310 human individuals and 7 tissues in 103 diverse strains of mice fed a normal chow or high-fat/high-sucrose (HFHS) diet. Variation of genes such as *FGF21*, *ADIPOQ*, *GCG*, and *IL6* showed enrichments which recapitulate experimental observations. Further, similar analyses were applied to explore both within-tissue signaling mechanisms (liver *PCSK9*) and genes encoding enzymes producing metabolites (adipose *PNPLA2*), where inter-individual correlation structure aligned with known roles for these critical metabolic pathways.

Examination of sex hormone receptor correlations in mice highlighted the difference of tissue-specific variation in relationships with metabolic traits. We refer to this resource as gene-derived correlations across tissues (GD-CAT) where all tools and data are built into a web portal enabling users to perform these analyses without a single line of code (gdcat.org). This resource enables querying of any gene in any tissue to find correlated patterns of genes, cell types, pathways, and network architectures across metabolic organs.

## eLife assessment

This **important** paper provides web based interface for cross-tissue analysis of omics datasets from – so far – two different human populations, with **compelling** evidence that the tool can be used to make meaningful scientific discoveries. Conceptually, these analyses are relevant for any systems biologist or bioinformatician who is interested in integrating large population datasets. Currently, the resource is already of use for scientists studying the HMDP or using GTEx data, and we hope to see updates in the coming years that incorporate more populations and more datatypes, which could make it a general tool for a wide community.

## Introduction

Interaction and/or coordination between organs is central to maintaining physiologic homeostasis among multicellular organisms. Beginning with the discovery of insulin over a century ago, characterization of molecules responsible for signal between tissues has required careful and elegant experimentation where these observations have been integral to deciphering physiology and disease. Further, actions of these molecules have been the key focus for development of potent therapeutics. For example, physiologic dissection of the actions of soluble proteins such as proprotein convertase subtilisin/kexin type 9 (*PCSK9*) and glucagon-like peptide 1 (*GLP1*) have yielded among the most promising therapeutics to treat cardiovascular disease and obesity, respectively (*Drucker, 2022*; *Trapp and Stanford, 2022*; *Dadu and Ballantyne, 2014*; *Lambert et al., 2012*). A majority of our understanding in how organs and cells utilize these mechanisms of tissue communication has arisen from elegant biochemical and physiologic experimentation. While these targeted investigations exist as the most definitive way to demonstrate causality for mechanisms, scaling such approaches to deconvolute the actions of tens of thousands of unique molecules which circulate in the blood becomes an impossible task. A major obstacle in the characterization of such soluble factors is that defining their tissues and pathways of action requires extensive experimental testing in cells and animal models.

Recent technological advances have enabled more unbiased views of molecules in circulation. Next-generation technologies have quantified thousands of factors in the blood across large populations. For example, large-scale proteomic measures have prioritized disease biomarkers and suggested involvements in genome-wide association mechanisms (*Anderson and Anderson, 2002*; *Ferkingstad et al., 2021*; *Sun et al., 2018*). Similar studies focused on integrating genetic variation with metabolomics quantification have yielded similar insights (*Nicholson et al., 2011*; *Kettunen et al., 2016*; *Harshfield et al., 2021*). However, the challenge is understanding which organs are secreting these molecules, how fast they are produced/degraded, and also what are the recipient tissues processing and/or responding to these factors. Furthermore, it is important to also identify the receptors that sense the secreted factor and enable the target organ to respond. This is challenging because the abundance of secreted factors and target receptors are dynamic, and rapidly change throughout the day or in response to a variety of environmental changes (e.g. diet or time of day). In addition, it is well known that genetic- or sex-driven variation can also modulate endocrine signaling. Hence, the foundations of therapeutic discovery require a comprehensive understanding of the mechanisms of endocrine signaling and here lies massive potential and an unmet need.

Previous studies in mouse and human populations have demonstrated that, when sufficiently powered, several known and new mechanisms of organ communication can be identified through simple global analyses of gene-gene correlations (*Seldin et al., 2018*; *Koplev et al., 2022*; *Seldin and Lusis, 2019*; *Cao et al., 2022*; *Velez et al., 2022*). The intuition behind this approach is that correlations across tissues and individuals will show a relatively normal distribution and upper-limit skews reflecting highly significant relationships have the potential to capture direct signaling. In this

study, we expanded on this intuition and tested the paralleled hypothesis that potential functions of signaling between tissues could be prioritized by focusing correlation analyses across individuals on specific genes. We highlight several areas where this approach was sufficient, as well as lacking in ability to recapitulate known tissue communication mechanisms. These analyses are contextualized by additional explorations of pathway-specific relationships (e.g. between go terms) and an example for context-specific gene-trait relationships for hormone receptors. In addition, we provide a user-friendly web tool to query these analyses in mouse and human population datasets at gdcat.org.

## Results

### Construction of a web tool to survey transcript correlations across tissues and individuals (GD-CAT)

Previous studies have established that 'brute force' analyses of correlation structure across tissues from population expression data can identify several known and new mechanisms of organ cross-talk. These were accomplished by surveying the global co-expression structure of all genes, where high correlation outliers highlighted proteins which elicit signaling (*Seldin et al., 2018*; *Koplev et al., 2022*; *Seldin and Lusis, 2019*; *Cao et al., 2022*; *Velez et al., 2022*). Following this intuition, we hypothesized that a paralleled but alternative approach to inter-individual correlation structure could be exploited to understand the functional consequences of specific genes. Our initial goal was to establish a user-friendly interface where all of these analyses and gene-centric queries could be performed without running any code. To accomplish this, we assembled a complete analysis pipeline (*Figure 1A*) as a shiny app and docker image hosted in a freely-available web address (gdcat.org). Here, users can readily-search gene correlation structure between individuals from filtered human (gene-by-tissue expression project [GTEx]) and mouse (hybrid mouse diversity panel [HMDP]) across tissues. GTEx is presently the most comprehensive pan-tissue dataset in humans (*Battle et al., 2017*), which was filtered for individuals where most metabolic tissues were sequenced (*Velez et al., 2022*). Collectively, this dataset contains 310 individuals, consisting of 210 male and 100 female (self-reported) subjects between the ages of 20–79. Data from the HMDP consisted of 96 diverse mouse strains fed a normal chow (5 tissues) or high-fat/high-sucrose (HFHS) diet (7 tissues) as well as carefully characterized clinical traits (*Parks et al., 2015*; *Hui et al., 2015*; *Norheim et al., 2021*; *Org et al., 2015*; *Lusis et al., 2016*; *Bennett et al., 2010*). Users first select a given species, followed by reported sex or diet (mouse) which loads the specified environment. Subsequent downstream analyses are then implemented accordingly from a specific gene in a given tissue. This selection prompts individual gene correlations across all other gene-tissue combinations using biweight midcorrelation (*Langfelder and Horvath, 2008*). From these charts, users are able to select a given tissue, where gene set enrichment analysis testing using clusterprofiler (*Yu et al., 2012*) and enrichR (*Kuleshov et al., 2016*) are applied to the correlated set of genes to determine the positively (activated) and negatively (suppressed) pathways which occur in each tissue. In addition to general queries of gene ~ gene correlation structure, comparison of expression changes is also visualized between age groups as well as reported sexes. In addition, we included the top cell-type abundance correlations with each gene. To compute cell abundance estimates from the same individuals, we used single-nucleus RNA-seq available from GTEx (*Jones et al., 2022*) and applied cellular deconvolution methods to the bulk RNA-seq (*Danziger et al., 2019*) (Materials and methods). Comparison of deconvolution methods (*Danziger et al., 2019*) showed that DeconRNA-Seq (*Gong and Szustakowski, 2013*) captured the most cell types within several tissues (*Figure 1—figure supplements 1–3*) and therefore was applied to all tissues where sn-RNA-seq was available. We note that visceral adipose, subcutaneous adipose, aortic artery, coronary artery, transverse colon, sigmoid colon, the heart left ventricle, the kidney cortex, liver, lung, skeletal muscle, spleen, and small intestine are the only tissues where sn-seq is available and not other tissues, such as brain, stomach, and thyroid.

We initially examined pan-tissue transcript correlation structures for several well-established mechanisms of tissue cross-talk via secreted proteins which contribute to metabolic homeostasis. Here, binning of the significant tissues and pathways related to each of these established secreted proteins resembled their known mechanisms of action (*Figure 1B–E*). For example, variation with subcutaneous adipose expression of *ADIPOQ* was enriched with genes in several metabolic tissues where it has been known to act (*Figure 1B*, left). In particular, subcutaneous adipose *ADIPOQ* expression

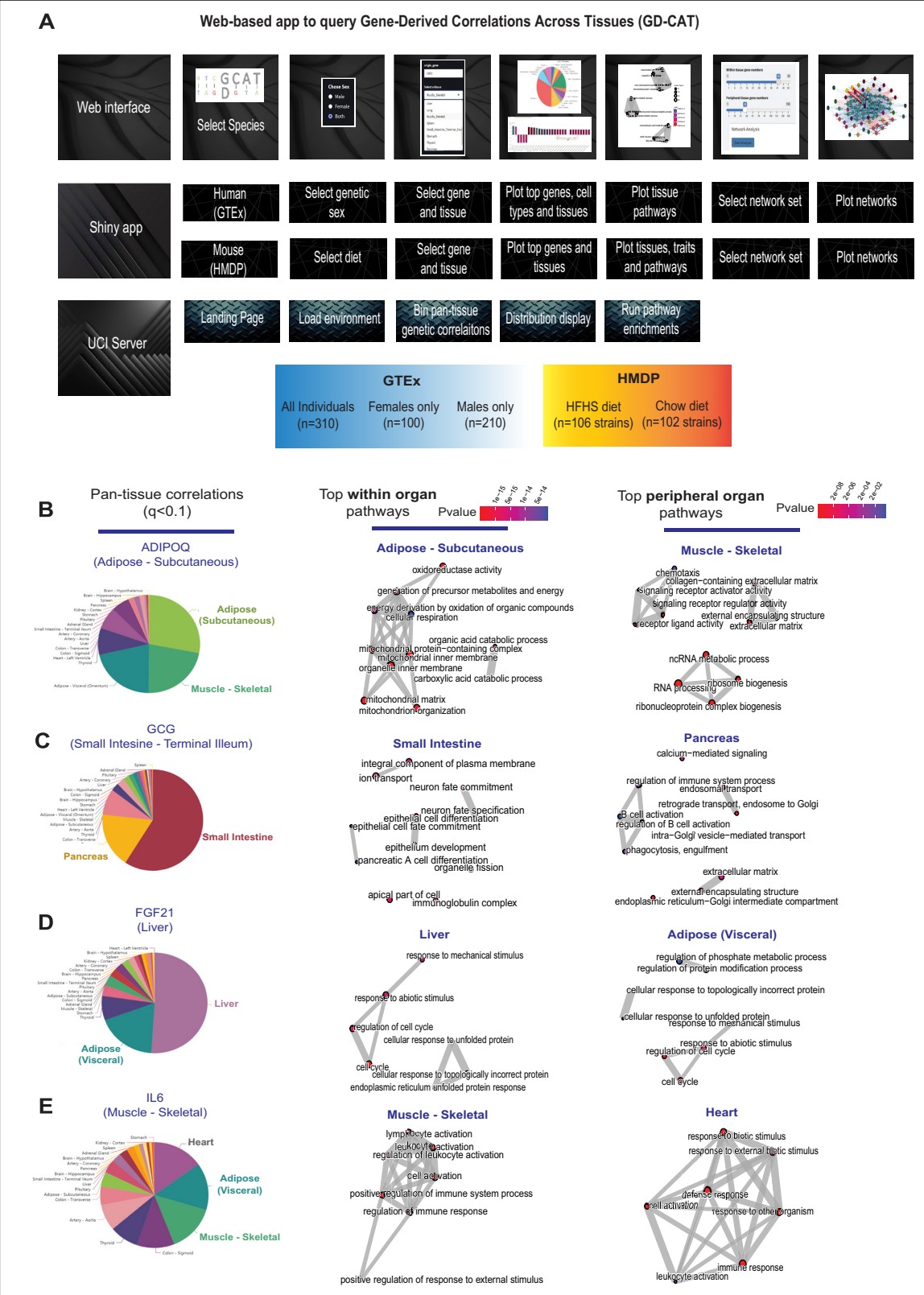

**Figure 1.** Web tool overview and inter-individual correlation structure of established endocrine proteins. (**A**) Web server structure for user-defined interactions, as well as server and shiny app implementation scheme for gene-derived correlations across tissues (GD-CAT). (**B**) All genes across the 18 metabolic tissues in 310 individuals were correlated with expression of *ADIPOQ* in subcutaneous adipose tissue, where a q-value cutoff of q<0.1 showed the strongest enrichments with subcutaneous and muscle gene expression (pie chart, left). Gene set enrichment analysis (GSEA) was performed

*Figure 1 continued on next page*

*Figure 1 continued*

using the bicor coefficient of all genes to *ADIPOQ* using gene ontology biological process annotations and network construction of top pathways using clusterprofiler, where pathways related to fatty acid oxidation were observed in adipose (left) and chemotaxis/ECM remodeling in skeletal muscle (right). (**B–D**) The same q-value binning, top within-tissue and top peripheral enrichments were applied to intestinal *GCG* (**C**), liver *FGF21* (**D**), and muscle *IL6* (**E**). For these analyses all 310 individuals (across both sexes) were used and q-value adjustments calculated using a Benjamini-Hochberg FDR adjustment.

The online version of this article includes the following figure supplement(s) for figure 1:

**Figure supplement 1.** Performance across four methods of cell-type deconvolution where relative proportions of cells (y-axis) are shown for all cell types annotated in single-cell reference (x-axis) in liver.

**Figure supplement 2.** Performance across four methods of cell-type deconvolution where relative proportions of cells (y-axis) are shown for all cell types annotated in single-cell reference (x-axis) in heart.

**Figure supplement 3.** Performance across four methods of cell-type deconvolution where relative proportions of cells (y-axis) are shown for all cell types annotated in single-cell reference (x-axis) in skeletal muscle.

**Figure supplement 4.** Pancreatic *INS* expression correlations across tissues in gene-by-tissue expression project (GTEx) were binned according to q<0.1 (top) and corresponding pancreatic gene set enrichment analysis (GSEA) network graph is shown (bottom).

correlated with fatty acid oxidative process within adipose (***Figure 1B***, middle) and was enriched with ECM, chemotaxis, and ribosomal biogenesis in skeletal muscle (***Figure 1B***, right). These correlated pathways align with the established physiologic roles of the protein in that fat-secreted adiponectin when oxidation is stimulated (***da Silva Rosa et al., 2021***; ***Straub and Scherer, 2019***) and muscle is a major site of action (***Ruan and Dong, 2016***). Beyond adiponectin, inter-individual correlation structure additionally recapitulated broad signaling mechanisms for other relevant endocrine proteins. For example, intestinal *GCG* (encoding GLP1, ***Figure 1C***), liver *FGF21* (***Figure 1D***), and skeletal muscle *IL6* (***Figure 1E***) showed binning patterns and pathway enrichments related to their known functions in pancreas (***Drucker, 2022***; ***McLean et al., 2021***), adipose tissue (***Fisher and Maratos-Flier, 2016***; ***Flippo and Potthoff, 2021***), and other metabolic organs (***Pedersen and Febbraio, 2008***), respectively. These analyses and web tool show some examples of exploring transcriptional correlation structure to confirm and identify mechanisms of signaling, where we note that additional limitations should be considered.

## Pathway-based examination of gene correlation structure and significance thresholds across tissues

While the select observations shown in ***Figure 1*** provide examples of support in exploring correlation structure of genes across inter-individual differences to investigate endocrinology, several limitations in these analyses should be considered. First, an additional explanation for a given gene showing strong correlation between the tissues could arise from a general pattern of correlation between the two tissues and not necessarily due to the discrete signaling mechanisms. In previous studies surveying correlation structure and network model architectures in the HMDP and STARNET populations, genes appeared generally stronger correlated between liver and adipose tissue compared to all other organ combinations explored (***Seldin et al., 2018***; ***Koplev et al., 2022***; ***Seldin and Lusis, 2019***). To investigate this global pattern of gene correlation structure between metabolic organs, we selected key gene ontology (GO) terms, KEGG pathways, and randomly sampled equal numbers of genes and evaluated relative significance of inter-tissue correlations across multiple statistical thresholds. These analyses suggested that usage of empirical Student's correlation p-values recapitulated a clear pattern of inter-tissue correlations between pathways (***Figure 2***). For example, comparison of the number of genes achieving significance of correlation between tissues among select GO terms revealed that tissues such as adipose and muscle appeared more correlated than spleen and other tissues at p-values less than 1e-3 (***Figure 2A***, left column). These global patterns of gene correlation between tissues among select pathways were reduced when the p-value threshold was lowered to 1e-6 (***Figure 2A***, middle column) or q-value adjustments (Materials and methods) were performed (***Figure 2A***, right two columns). For these reasons, only q-value adjusted value was used and implemented into pie charts providing the tissue-specific occurrences of correlated genes at three thresholds (q<0.1, q<0.01, q<0.001) within the web tool. Next, in order to further evaluate these global patterns of innate transcript correlation structure and determine whether they reflected concordance

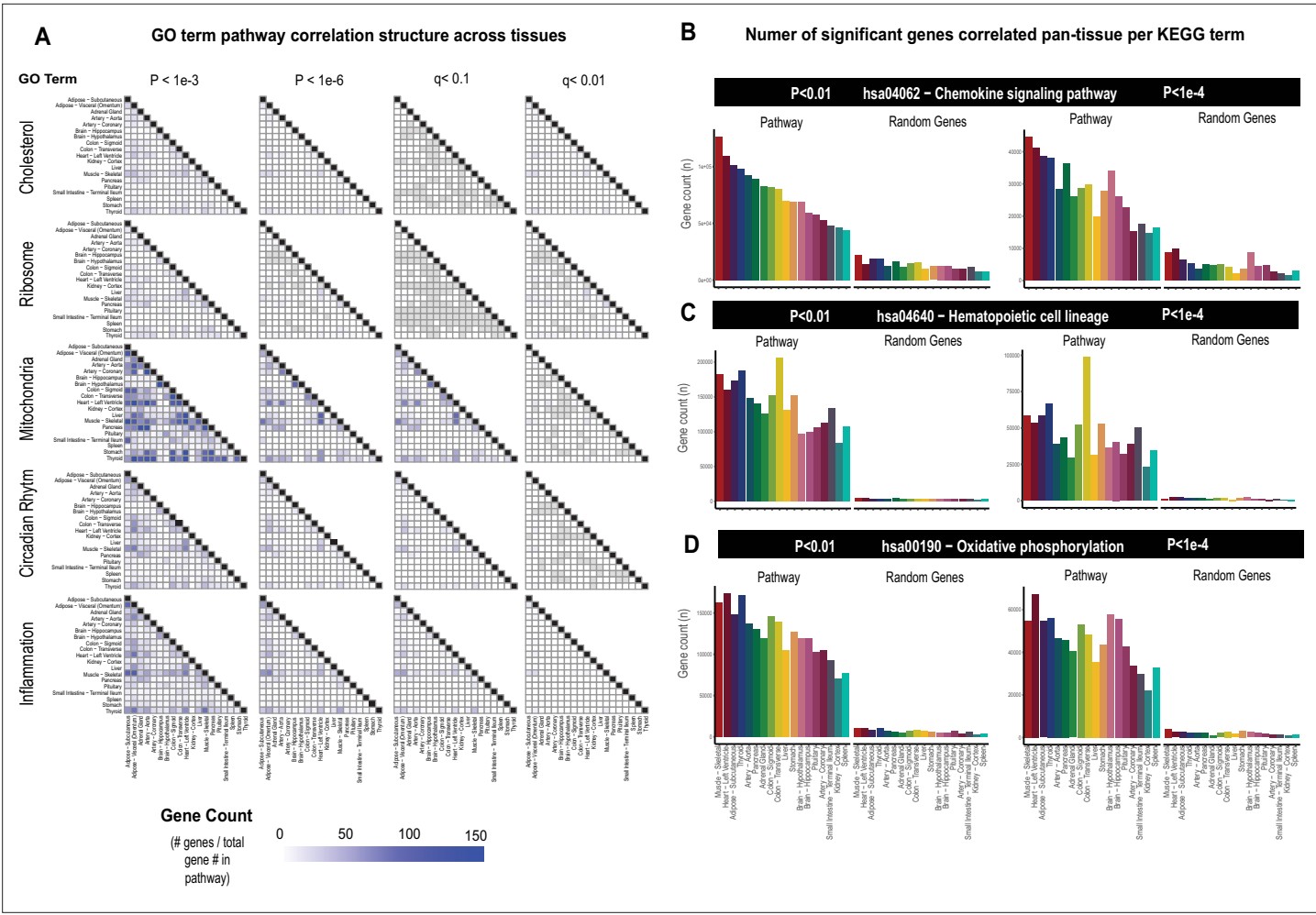

**Figure 2.** Tissue-specific contributions to pan-organ gene-gene correlation structure. (**A**) Heatmap showing all the number of gene-gene correlations across tissues which achieve significance relative to total number of genes in each pathway at biweigth midcorrelation Student's p-value <1e-3 (left column), p-value <1e-6 (left middle column) of BH-corrected q-value <0.1 (right middle column) or BH-corrected q-value <0.01 (right column). Within-tissue correlations are omitted from this analysis. (**B–D**) Genes corresponding to each KEGG pathway shown were correlated both within and across all other organs where the number of genes which meet each Student's p-value threshold are shown (y-axis). Tissues (x-axis) are rank-ordered by the number of genes which correlate for hsa04062 – chemokine signaling pathway at p-value <0.01 and shown for other KEGG terms, hsa04640 – hematopoietic cell lineage (**C**) and hsa00190 – oxidative phosphorylation (**D**) and additionally p-value <1e-4 (right side).

between known metabolic pathways or innate to the dataset used, tissues were rank-ordered by the number of genes which meet p-value thresholds and compared to randomly sampled genes of similar pathway sized (*Figure 2B*). Among KEGG pathways selects (hsa04062 – chemokine signaling pathway, hsa04640 – hematopoietic cell lineage, and hsa00190 – oxidative phosphorylation), the top-ranked organs by correlated gene numbers differed (skeletal muscle, colon, and thyroid, respectively); however, a general trend of specific tissues ranking higher than others were observed (*Figure 2B*). For example, skeletal muscle and heart appeared among the strongest correlated across pathways and organs, compared to kidney cortex and spleen which were observed to rank among the lowest (*Figure 2B*, pathways). We note that when the same analysis was performed on randomly sampled genes from each organ consisting of the same number as genes within each KEGG pathway, these rankings and number of significant correlating genes were no longer observed (*Figure 2B*, random genes), suggesting that in certain instances differences between organs in general connectivity to others might reflect concordance between known pathways. It is important to consider here that for the organs ranking lower, the lack of relative correlating numbers is likely due to sparsity of available data and not necessarily general patterns of gene correlation. This point is supported by the fact that among the lowest-ranked 33% of tissues across pathways, we observed a significant negative

overall correlation (bicor=–0.45, p-value=2.3e-5) between number of NA values per individual and the gene count for significance shown in *Figure 2B*. This negative correlation between missing data and number of significant correlations for pathways across tissues was not observed when binning the top 33% (bicor=0.09, p-value=0.42) or middle 33% (bicor=–0.12, p-value=0.27) of organs. Collectively, these analyses show that innate correlation structures exist between organs which differ depending on pathways investigated and that tissues which don't show broad correlation structure could potentially be attributed to areas of missing data among GTEx.

## PCSK9 signaling and lipid exchange between adipose and muscle apparent in simple network models of correlation structure

Next, we wanted to ask whether our approach of analyzing inter-individual correlation structure across tissue for endocrine proteins was also sufficient to define within-tissue signaling mechanisms or actions of enzymes producing metabolites that signal across organs. Dissimilar to the cross-tissue distributions of significance in *Figure 1*, the same analysis of liver *PCSK9* highlighted exclusively liver genes which were varied together (*Figure 3A*), in particular those involved in cholesterol metabolism/ homeostasis (*Figure 3B*). Consistent with the established role for PCSK9 as a primary degradation mechanism of LDLR (*Lambert et al., 2012*; *Peterson et al., 2008*), network model construction of correlated genes highlighted the gene as a central node linking cholesterol biosynthetic pathways with those involved in other metabolic pathways such as insulin signaling (*Figure 3C*). Given that organ signaling via metabolites comprises many critical processes among multicellular organisms, our next goal was to apply this gene-centric analyses to established mechanisms of metabolite signaling. The gene *PNPLA2* encodes adipose triglyceride lipase which localizes to lipid droplets and breaks down triglycerides for oxidation or mobilization as free fatty acids for peripheral tissues (*Zechner et al., 2009*). Variation in expression of *PNPLA2* showed highly significant enrichments with beta oxidation pathways in adipose tissue (*Figure 3D*). Muscle pathways enriched for the gene were represented by sarcomere organization and muscle contraction (*Figure 3F*). Construction of an undirected network from these expression data placed the gene as a central node between the two tissues, linking regulators of adipose oxidation (*Figure 3F*, red) to muscle contractile process (*Figure 3F*, purple) where additional strongly co-correlated genes were implicated as additional candidates (*Figure 3F*). In sum, these analyses provide two examples of within-liver signaling via *PCSK9* and adipose-muscle communication through *PNPLA2* where the top-correlated genes and network models recapitulate known mechanisms. Given the utility of these undirected network models, a function in gene-derived correlations across tissues (GD-CAT) was added to enable users to generate network models for any gene-tissue combination and select parameters such as number of within-tissue and peripheral correlated genes to include.

## Inter-individual correlation analysis of HMDP highlights tissue- and diet-specific phenotype relationships with sex hormones

Genetic reference panels in model organisms, such as mice, present appeal in studying complex traits in that environmental conditions can be tightly controlled, tissues and invasive traits readily accessible, and the same, often renewable, genetic background can be studied and compared among multiple exposures such as diets or drug treatments (*Lusis et al., 2016*; *Li and Auwerx, 2020*; *Andreux et al., 2012*; *Seldin et al., 2019*). For this resource, we utilized data from the HMDP fed a normal chow (*Lusis et al., 2016*; *Bennett et al., 2010*) or HFHS diet for 8 weeks (*Parks et al., 2015*; *Hui et al., 2015*; *Norheim et al., 2021*; *Org et al., 2015*). While the number of tissues available was less than in GTEx, these panels allow for comparison of how gene correlations shift depending on diet. Therefore, queries of gene correlation in mice were segregated into either chow or HFHS diet and an additional panel to download a table or visualize the relationship between genes and clinical measures was added. The inferred abundances of cell types from each individual are correlated across user-defined genes, with the bicor coefficient plotted for each cell type.

One advantage of HMDP data compared to GTEx is the abundance of phenotypic measures available within each cohort. To show the utility of examining correlations within this reference panel, we selected sex hormone receptors androgen receptor (*Ar*), estrogen receptor alpha (*Esr1*), or estrogen receptor beta (*Esr2*) and binned the top 10 phenotypes which were correlated. These analyses were stratified based on where sex hormones were expressed (either liver or adipose tissue) or dietary

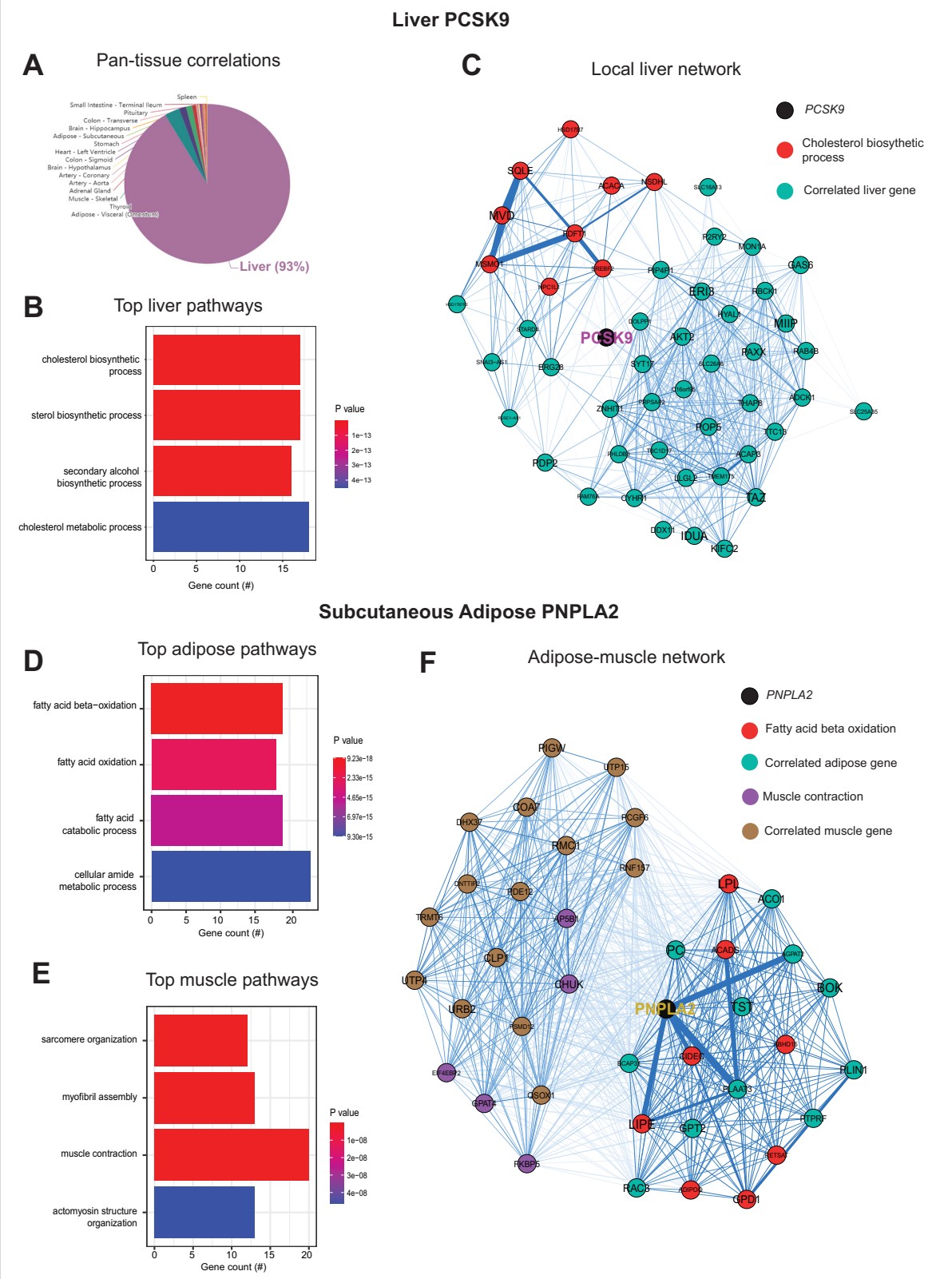

**Figure 3.** Inter-individual transcript correlation structure and network architecture of liver *PCSK9* and adipose *PNPLA2*. (**A**) Distribution of pan-tissue genes correlated with liver *PCSK9* expression (q<0.1), where 93% of genes were within liver (purple). (**B**) Gene ontology (GO) (BP) overrepresentation test for the top 500 hepatic genes correlated with *PCSK9* expression in liver. (**C**) Undirected network constructed from liver genes (aqua) correlated with *PCSK9*, where those annotated for 'cholesterol biosythetic process' are colored in red. (**D–E**) Overrepresentation tests corresponding to the

*Figure 3 continued on next page*

*Figure 3 continued*

top-correlated genes with adipose (subcutaneous) *PNPLA2* expression residing in adipose (**D**) or peripherally in skeletal muscle (**E**). (**F**) Undirected network constructed from the strongest correlated subcutaneous adipose tissue (light aqua) and muscle genes (light brown) with PNPLA2 (black), where genes corresponding to GO terms annotated as 'fatty acid beta oxidation' or 'muscle contraction' are colored purple or red, respectively. For these analyses all 310 individuals (across both sexes) were used and q-value adjustments calculated using a Benjamini-Hochberg FDR adjustment. Network graphs generated based in biweight midcorrelation coefficients, where edges are colored blue for positive correlations or red for negative correlations. Network edges represent positive (blue) and negative (red) correlations and the thicknesses are determined by coefficients. They are set for a range of bicor=0.6 (minimum to include) to bicor=0.99.

regiment of the ~100 strains (normal chow or HFHS diet). This analysis demonstrated the difference in relationships between tissue location of sex hormone receptor and dietary context with metabolic traits. For example, expression of *Ar* in adipose tissue among HMDP mice fed an HFHS diet was negatively correlated with fat mass and body weight traits, whereas expression in liver oppositely correlated with the same traits in a positive direction (*Figure 4A*). The top traits which correlated also differed by tissue or expression for *Ar*, such as plasma lipid parameters in adipose tissue compared to blood cell traits in chow-fed mice (*Figure 4A*). We note that among the three hormone receptors

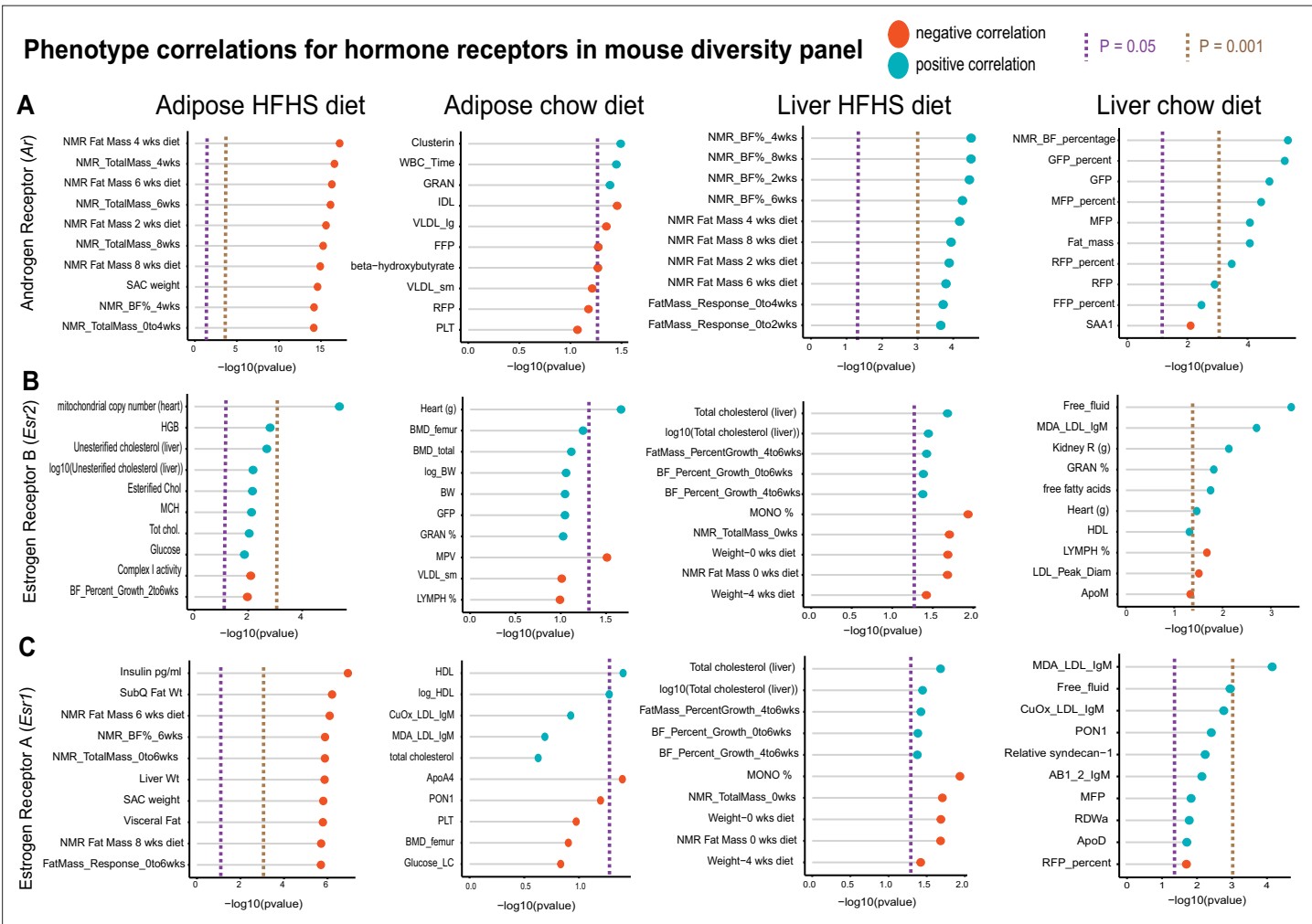

**Figure 4.** Hybrid mouse diversity panel (HMDP) tissue- and diet-specific correlations of sex hormone receptors. The top 10 phenotypic traits are shown for correlations with expression of androgen receptor (**A**), estrogen receptor 1 (**B**), or estrogen receptor 2 (**C**), colored by direction in the hybrid mouse diversity panel. Positive correlations are shown in light blue and negative correlations as sunset orange, where phenotypes (y-axis) are ordered by significance (x-axis, -log10(p-value) of correlation). Correlations are segregated by whether sex hormone receptors are expressed by gonadal adipose tissue (left two columns) in ~100 HMDP strains fed a high-fat/high-sucrose (HFHS) diet (left), normal chow diet (left middle), or liver-expressed receptors fed an HFHS diet (right middle) or normal chow diet (right). Dashed lines show a Student's correlation p-value (from bicor) of p=0.05 (purple) or p=0.001 (beige).

investigated, *Esr2* appeared the most consistently correlated between tissues and diets with metabolic traits (*Figure 4B*). Expression of *Esr1* also showed a clear tissue and diet difference in the traits which were the most strongly co-regulated. Under HFHS dietary conditions, a negative correlation with insulin and fat pad weights were observed exclusively with adipose expression, while positive correlations with liver lipids were observed with expression in liver (*Figure 4C*). These analyses highlight how phenotype correlations in mouse populations can help to determine contexts relevant for gene regulation and point to the diversity of potential contexts relevant for sex hormone receptors in metabolic tissues.

## Discussion
### Limitations and conclusions

Here, we provide a new resource to explore correlations across organ gene expression in the context of inter-individual differences. We highlight areas where these align with established and relevant mechanisms of physiology and suggest that similar explorations could be used as a discovery tool. Several key limitations should be considered when exploring GD-CAT for mechanisms of inter-tissue signaling though. Primarily, the fact that correlation-based analyses could reflect both causal and reactive patterns of variation. While several statistical methods such as mediation (*Richiardi et al., 2013*; *Zeng et al., 2021*) and Mendelian randomization (*Emdin et al., 2017*; *Sanderson et al., 2022*) exist to further refine causal inferences, likely the only definitive method to distinguish is in carefully designed experimentation. Further, analyses of genetic correlation (e.g. correlations considering genetic loci to infer causality) also present appeal in refining some causal mechanisms. Correlation between molecular and phenotypic variables can occur for a variety of reasons, not just between their individual relationships, but often more broadly, from a variety of complex genetic and environmental factors. Further, many correlations tend to be dominated by genes expressed within the same organ. This could be due to the fact that, within-tissue correlations could capture both the pathways regulating expression of a gene and potential consequences of changes in expression/function, and distinguishing between the two presents a significant challenge. For example, a GD-CAT query of insulin (*INS*) expression in pancreas shows exclusive enrichments in pancreas and corresponding pathway terms reflect regulatory mechanisms such as secretion and ion transport (*Figure 1—figure supplement 4*). Representation of given genes may also differ significantly depending on the dataset used. For example, queries of other tissues correlated with the critical X inactive specific transcript (*XIST*), in liver show no significant correlations at qvalue cutoffs used. This is due to the fact that the gene operates in a sex-dependent manner, where females are significantly less represented in GTEx and liver exists as a sparser tissue compared to others (*Figure 2*). In addition, the analyses presented are derived from differences in gene expression across individuals which arise from complex interaction of genetic and environmental variables. Expression of a gene and its corresponding protein can show substantial discordances depending on the dataset used. These have been discussed in detail (*Liu et al., 2016*; *Maier et al., 2009*; *Buccitelli and Selbach, 2020*), but ranges of co-correlation can vary widely depending on the datasets used and approaches taken. We note that for genes encoding proteins where actions from acute secretion grossly outweigh patterns of gene expression, such as insulin, caution should be taken when interpreting results. As the depth and availability of tissue-specific proteomic levels across diverse individuals continues to increase, an exciting opportunity is presented to explore the applicability of these analyses and identify areas when gene expression is not a sufficient measure. For example, mass-spec proteomics was recently performed on GTEx (*Jiang et al., 2020*); however, given that these data represent 6 individuals, analyses utilizing well-powered inter-individual correlations such as ours which contain 310 individuals remain limited in applications.

The queries provided in GD-CAT use fairly simple linear models to infer organ-organ signaling; however, more sophisticated methods can also be applied in an informative fashion. For example, Koplev et al. generated co-expression modules from nine tissues in the STARNET dataset, where construction of a massive Bayesian network uncovered interactions between correlated modules (*Koplev et al., 2022*). These approaches expanded on analysis of STAGE data to construct network models using WGCNA across tissues and relating these resulting eigenvectors to outcomes (*Talukdar et al., 2016*). The generalized approach of constructing cross-tissue gene regulatory modules presents appeal in that genes are able to be viewed in the context of a network with respect to all

other gene-tissue combinations. In searching through these types of expanded networks, individuals can identify where the most compelling global relationships occur. One challenge with this type of approach, however, is that co-regulated pathways and module members are highly subjective to parameters used to construct GRNs (e.g. reassignment threshold in WGCNA) and can be difficult in arriving at a 'ground truth' for parameter selection. We note that the WGCNA package is also implemented in these analyses, but solely to perform gene-focused correlations using biweight midcorrelation to limit outlier inflation. While the midweight bicorrelation approach to calculate correlations could also be replaced with more sophisticated models, one consideration would be a concern of overfitting models and thus, biasing outcomes.

In another notable example MultiCens was developed as a tool to uncover communication between genes and tissues and applied to suggest central processes which exist in multi-layered data relevant for Alzheimer's disease (*Kumar et al., 2022*). In addition, Jadhav and colleagues adopted a machine learning approach to mine published literature for relationships between hormones and genes (*Jadhav et al., 2022*). Further, association mapping of plasma proteomics data has been extensively applied and intersection with genome-wide association disease loci has offered intriguing potential disease mechanisms (*Ferkingstad et al., 2021*; *Suhre et al., 2021*). Another common application to single-cell sequencing data is to search for overrepresentation of known ligand-receptor pairs between cell types (*Armingol et al., 2021*). These and additional applications to explore tissue communication/coordination present unique strengths and caveats, depending on the specific usage desired. Regardless of methods used to decipher, one important limitation to consider in all these analyses is the nature of underlying data. For example, our evaluation of GTEx data structure suggested that important organs such as spleen and kidney were insufficient due to availability in matching expression data between individuals. Further, GTEx sample varies as to the collection times, sample processing times, and other important parameters such as cause of death. Mouse population data such as the HMDP or BxD cohorts offer appeal in these regards, as environmental conditions and collection times are easily fixed. Regardless, careful consideration of how data was generated and normalized are fundamental to interpreting results.

In sum, we demonstrate that adopting a gene-centric approach to surveying correlation structure of transcripts across organs and individuals can inform mechanism of coordination between metabolic tissues. Initially, we queried several well-established and key mediators of physiologic homeostasis, such as *FGF21, GCG,* and *PCSK9*. These approaches are further suggested to be applicable to mechanisms of metabolite signaling, as evident by pan-tissue investigation of adipose *PNPLA2*. Exploration of HMDP data highlighted the diverse phenotype correlations depending on tissue and diet for sex hormone receptors. To facilitate widespread access and use of this transcript isoform-centric analysis of inter-individual correlations, a full suite of analyses such as those performed here can be performed from a lab-hosted server (gdcat.org) or in isolation from a shiny app or docker image.

## Materials and methods
### Availability of web tool and analyses

All analyses, datasets, and scripts used to generate the associated web tool (GD-CAT) can be accessed via GitHub (copy archived at *Zhou and Seldin, 2023*) or within the associated docker image. In addition, access to the GD-CAT web tool is also available through the web portal gdcat.org. This portal was created to provide a user-friendly interface for accessing and using the GD-CAT tool without the need to download or install any software or packages. Users can simply visit the website, process data, and start using the tool. Corresponding tutorial and the other resources were made available to facilitate the utilization of the web tool on GitHub. The interface and server of the web were built and linked based on the shiny package using R (v. 4.2.0). Shiny package provides a powerful tool for building interactive web applications using R, allowing for fast and flexible development of custom applications with minimal coding required.

### Pathway-specific gene correlations across tissues

Detailed scripts and analyses for pathway-specific investigations across tissues in *Figure 2* are provided in GitHub (copy archived at *Tamburini, 2023*). Briefly, to interrogate broad tissue correlation structure, the number of genes which passed each biweight midcorrelation p-value cutoff is shown

normalized to the total number of genes corresponding to that pathway term. Pathways were selected by accessing all available GO annotations for all genes using the Universal Protein Resource (*The UniProt Consortium, 2017*) and subsetting genes where a given term is listed. To determine which tissues show the most co-correlation across genes and organs, KEGG terms shown were selected and each corresponding gene-tissue combinations were correlated. Tissues were then binned individually by the number of significant correlations which were observed across organs among each selected KEGG pathway at indicated correlation p-values. Rank-ordering on the figure was shown by chemokine signaling at p<0.01 and each term was compared to a randomly sampled set of genes corresponding to the same number contained in each pathway.

## Data sources and availability

All human data used in this study can be immediately accessed via web tool or docker to facilitate analysis. Metabolic tissue data was accessed through GTEx V8 downloads portal on August 18, 2021, and previously described (*Velez et al., 2022*; *Battle et al., 2017*). These raw data can also be readily accessed from the associated R-based walkthrough via GitHub (copy archived at *Velez, 2022*). Briefly, these data were filtered to retain genes which were detected across tissues where individuals were required to show counts >0 across all data. Given that our goal was to look across tissues at enrichments, this was done to limit spurious influence of genes only expressed in specific tissues in specific individuals. HMDP data was collected from previously described studies (*Parks et al., 2015*; *Lusis et al., 2016*; *Bennett et al., 2010*; *Seldin et al., 2019*) and inter-individual differences were compared at the strain level to maximize possible comparisons between historical data.

## Correlation analyses across tissues

Biweight midcorrelation coefficients and corresponding p-values within and across tissues were generated using WGCNA bicorandpvalue() function (*Langfelder and Horvath, 2008*). We note that while the WGCNA package was used to calculate coefficients and corresponding Student's p-values, this generalized framework does not utilize any module generation. Associated q-value adjustments were applied using the Benjamini-Hochberg FDR from the R package 'stats'. The BH procedure was selected instead of other FDR control methods because of its efficiency in CPU usage on the hosted server.

## Pathway enrichment analyses

Pathway enrichments were generated using gene set enrichment analyses available from the R package clusterProfiler. Specifically, the bicor coefficients were used as the rank weight of each gene and enrichment tests performed by permuting against the human or mouse reference transcriptome. Terms used for the enrichment analyses were derived from GO (biological process, cellular component, and molecular function) which were accessed using the R package enrichR. For this analysis and on the available app, input genes were determined at indicated q-value threshold.

## Deconvolution of bulk tissue seq data on web tool

All scripts and deconvolution data produced are available at GitHub (copy archived at *Van, 2022*). Briefly, sn-RNA-seq data was accessed from the Human Cell Atlas (*Jones et al., 2022*) for matching organ datasets with metabolic tissues. From these data, four deconvolution methods were applied using ADAPTS (*Danziger et al., 2019*) where DeconRNA-Seq (*Gong and Szustakowski, 2013*) was selected for its ability to capture the abundances of the most cell types across tissues such as liver, heart, and skeletal muscle (*Figure 1—figure supplements 1–3*). The full combined matrix was assembled for DeconRNA-Seq results across individuals in GTEx where correlations between cell types and genes were performed also using the bicorandpvalue() in WGCNA (*Langfelder and Horvath, 2008*).

## Acknowledgements

We acknowledge the following funding sources for supporting these studies: MZ, CNF, IT, CMN, LMV, CV, CJ, and MMS were supported by NIH grants HL138193, DK130640, and DK097771. ALH is supported by NIH grants U54 DK120342, R01 DK109724, and P30 DK063491. NP is supported by NIH grant R37 CA266042. HB was supported by National Research Foundation of Korea (2021R1A6A3A14039132). JL was supported by an NIH grant F31DK134173-01A1. CJ was supported

by AASLD Foundation Pinnacle Research Award in Liver Disease, Edward Mallinckrodt, Jr. Foundation Award, and an NIH grant AA029124.

## Additional information

### Competing interests

Benjamin L Parker, Marcus M Seldin: Reviewing editor, *eLife*. The other authors declare that no competing interests exist.

### Funding

| Funder | Grant reference number | Author |
|---|---|---|
| National Institutes of Health | HL138193 | Mingqi Zhou<br>Carlos HV Nascimento-Filho<br>Ian Tamburini<br>Christy M Nguyen<br>Leandro M Velez<br>Cassandra Van<br>Cholsoon Jang<br>Marcus M Seldin |
| National Institutes of Health | DK130640 | Mingqi Zhou<br>Carlos HV Nascimento-Filho<br>Ian Tamburini<br>Christy M Nguyen<br>Leandro M Velez<br>Cassandra Van<br>Cholsoon Jang<br>Marcus M Seldin |
| National Institutes of Health | DK097771 | Mingqi Zhou<br>Carlos HV Nascimento-Filho<br>Ian Tamburini<br>Christy M Nguyen<br>Leandro M Velez<br>Cassandra Van<br>Cholsoon Jang<br>Marcus M Seldin |
| National Institutes of Health | DK120342 | Andrea L Hevener |
| National Institutes of Health | DK109724 | Andrea L Hevener |
| National Institutes of Health | DK063491 | Andrea L Hevener |
| National Institutes of Health | CA266042 | Nicholas Pannunzio |
| National Research Foundation of Korea | 2021R1A6A3A14039132 | Hosung Bae |
| National Institutes of Health | F31DK134173-01A1 | Johnny Le |
| AASLD Foundation | Pinnacle Research Award in Liver Disease | Cholsoon Jang |
| Edward Mallinckrodt, Jr Foundation | | Cholsoon Jang |
| National Institutes of Health | AA029124 | Cholsoon Jang |

| Funder | Grant reference number | Author |
|---|---|---|

The funders had no role in study design, data collection and interpretation, or the decision to submit the work for publication.

## Author contributions

Mingqi Zhou, Resources, Data curation, Software, Formal analysis, Validation, Investigation, Methodology, Writing – original draft, Writing – review and editing; Ian Tamburini, Software, Formal analysis, Visualization, Methodology, Writing – review and editing; Cassandra Van, Software, Formal analysis, Writing – original draft; Jeffrey Molendijk, Formal analysis, Writing – original draft, Writing – review and editing; Christy M Nguyen, Formal analysis, Investigation, Methodology, Writing – review and editing; Ivan Yao-Yi Chang, Resources, Software, Methodology, Project administration; Casey Johnson, Formal analysis, Funding acquisition, Visualization; Leandro M Velez, Resources, Data curation, Validation; Youngseo Cheon, Supervision, Visualization, Methodology; Reichelle Yeo, Formal analysis, Investigation, Writing – review and editing; Hosung Bae, Data curation, Formal analysis, Validation; Johnny Le, Ron Pulido, Resources, Software, Investigation; Natalie Larson, Resources, Software, Formal analysis; Carlos HV Nascimento-Filho, Validation, Investigation, Writing – original draft; Cholsoon Jang, Ivan Marazzi, Conceptualization, Funding acquisition, Validation; Jamie Justice, Validation, Investigation, Project administration; Nicholas Pannunzio, Validation, Visualization, Writing – original draft; Andrea L Hevener, Conceptualization, Data curation, Supervision, Funding acquisition, Validation, Writing – original draft; Lauren Sparks, Conceptualization, Supervision, Funding acquisition, Writing – original draft; Erin E Kershaw, Conceptualization, Funding acquisition, Validation, Writing – original draft; Dequina Nicholas, Conceptualization, Data curation, Validation, Visualization, Writing – original draft; Benjamin L Parker, Conceptualization, Visualization, Writing – original draft; Selma Masri, Conceptualization, Validation, Methodology; Marcus M Seldin, Conceptualization, Resources, Data curation, Software, Formal analysis, Supervision, Funding acquisition, Validation, Investigation, Writing – original draft, Writing – review and editing

## Author ORCIDs

Mingqi Zhou (ID) http://orcid.org/0009-0007-7643-7873
Jeffrey Molendijk (ID) http://orcid.org/0000-0001-6575-504X
Carlos HV Nascimento-Filho (ID) https://orcid.org/0000-0002-6870-2602
Jamie Justice (ID) http://orcid.org/0000-0003-2953-4404
Dequina Nicholas (ID) http://orcid.org/0000-0003-4996-2190
Benjamin L Parker (ID) http://orcid.org/0000-0003-1818-2183
Selma Masri (ID) http://orcid.org/0000-0002-8619-8331
Marcus M Seldin (ID) https://orcid.org/0000-0001-8026-4759

Reviewer #1 (Public Review): https://doi.org/10.7554/eLife.88863.3.sa1
Reviewer #2 (Public Review): https://doi.org/10.7554/eLife.88863.3.sa2
Author Response https://doi.org/10.7554/eLife.88863.3.sa3

# Additional files

## Supplementary files

• MDAR checklist

## Data availability

All previously published datasets are listed below. Please see the Materials and methods for further details (section 'Data sources and availability').

The following previously published datasets were used:

| Author(s) | Year | Dataset title | Dataset URL | Database and Identifier |
|---|---|---|---|---|
| Lonsdale J, Thomas J, Salvatore M, Phillips R, Lo E, Shad S, Hasz R, Walters G, Garcia F, Young N, Foster B, Moser M, Karasik E, Gillard B, Ramsey K, Sullivan S, Bridge J, Magazine H, Syron J, Fleming J, Siminoff L, Traino H, Mosavel M, Barker L, Jewell S, Rohrer D, Maxim D, Filkins D, Harbach P, Cortadillo E, Berghuis B, Turner L, Hudson E, Feenstra K, Sobin L, Robb J, Branton P, Korzeniewski G, Shive C, Tabor D, Qi L, Groch K, Nampally S, Buia S, Zimmerman A, Smith A, Burges R, Robinson K, Valentino K, Bradbury D, Cosentino M, Diaz-Mayoral N, Kennedy M, Engel T, Williams P, Erickson K, Ardlie K, Winckler W, Getz G, DeLuca D, MacArthur D, Kellis M, Thomson A, Young T, Gelfand E, Donovan M, Meng Y, Grant G, Mash D, Marcus Y, Basile M, Liu J, Zhu J, Tu Z, Cox NJ, Nicolae DL, Gamazon ER, Im HK, Konkashbaev A, Pritchard J, Stevens M, Flutre T, Wen X, Dermitzakis ET, Lappalainen T, Guigo R, Monlong J, Sammeth M, Koller D, Battle A, Mostafavi S, McCarthy M, Rivas M, Maller J, Rusyn I, Nobel A, Wright F, Shabalin A, Feolo M, Sharopova N, Sturcke A, Paschal J, Anderson JM, Wilder EL, Derr LK, Green ED, Struewing JP, Temple G, Volpi S, Boyer JT, Thomson EJ, Guyer MS, Ng C, Abdallah A, Colantuoni D, Insel TR, Koester SE, Little AR, Bender PK, Lehner T, Yao Y, Compton CC, Vaught JB, Sawyer S, Lockhart NC, Demchok J, Moore HF | 2013 | Adult Genotype-Tissue Expression (GTEx) project | https://gtexportal.org/home/downloads/adult-gtex | GTExPortal, adult-gtex |

*Continued on next page*

*Continued*

| Author(s) | Year | Dataset title | Dataset URL | Database and Identifier |
|---|---|---|---|---|
| Farber CR, Bennett BJ, Orozco L, Eskin E, Lusis AJ | 2011 | Bone mineral density in males of 96 strains of mice in the Hybrid Mouse Diversity Panel (HMDP) | https://phenome.jax.org/projects/HMDPpheno1 | Mouse Phenome Database, HMDPpheno1 |
| Ghazalpour A, Bennett BJ, Orozco L, Lusis AJ | 2014 | Liver metabolite levels: Nucleotides and peptides in males of 104 strains of the Hybrid Mouse Diversity Panel (HMDP) | https://phenome.jax.org/projects/HMDPpheno10 | Mouse Phenome Database, HMDPpheno10 |
| Bennett BJ, Lusis AJ | 2015 | Diet effects on aortic lesion size in 103 Hybrid Mouse Diversity Panel strains that are transgenic for human APOE*3Leiden and human CETP (F1 hybrids) | https://phenome.jax.org/projects/HMDPpheno11 | Mouse Phenome Database, HMDPpheno11 |
| Bennett BJ, Farber CR, Orozco L, Eskin E, Lusis AJ | 2010 | Plasma lipids and body composition in males of 99 strains of mice in the Hybrid Mouse Diversity Panel (HMDP) | https://phenome.jax.org/projects/HMDPpheno2 | Mouse Phenome Database, HMDPpheno2 |
| Korshunov VA | 2012 | Heart rate and blood pressure in males of 58 strains of mice | https://phenome.jax.org/projects/HMDPpheno3 | Mouse Phenome Database, HMDPpheno3 |
| Park CC, Gale GD, Lusis AJ, Smith DJ | 2011 | Fear conditioning in males of 94 strains of mice in the Hybrid Mouse Diversity Panel (HMDP) | https://phenome.jax.org/projects/HMDPpheno4 | Mouse Phenome Database, HMDPpheno4 |
| Davis RC, van Nas A, Bennett BJ, Eskin E, Lusis AJ | 2013 | Hematological parameters in males of 83 strains of mice in the Hybrid Mouse Diversity Panel (HMDP) | https://phenome.jax.org/projects/HMDPpheno5 | Mouse Phenome Database, HMDPpheno5 |
| Ghazalpour A, Bennett BJ, Orozco L, Lusis AJ | 2014 | Liver metabolite levels: Amino acids in males of 104 strains of the Hybrid Mouse Diversity Panel (HMDP) | https://phenome.jax.org/projects/HMDPpheno6 | Mouse Phenome Database, HMDPpheno6 |
| Ghazalpour A, Bennett BJ, Orozco L, Lusis AJ | 2014 | Liver metabolite levels: Carbohydrates, cofactors and vitamins, energy, and xenobiotics in males of 104 strains of the Hybrid Mouse Diversity Panel (HMDP) | https://phenome.jax.org/projects/HMDPpheno7 | Mouse Phenome Database, HMDPpheno7 |
| Ghazalpour A, Bennett BJ, Orozco L, Lusis AJ | 2014 | Liver metabolite levels: Lipids (except lysolipids) in males of 104 strains of the Hybrid Mouse Diversity Panel (HMDP) | https://phenome.jax.org/projects/HMDPpheno8 | Mouse Phenome Database, HMDPpheno8 |
| Ghazalpour A, Bennett BJ, Orozco L, Lusis AJ | 2014 | Liver metabolite levels: Lipids (lysolipids) in males of 104 strains of the Hybrid Mouse Diversity Panel (HMDP) | https://phenome.jax.org/projects/HMDPpheno9 | Mouse Phenome Database, HMDPpheno9 |

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
