## [Editor Report · eLife assessment]

This **important** paper provides web based interface for cross-tissue analysis of omics datasets from – so far – two different human populations, with **compelling** evidence that the tool can be used to make meaningful scientific discoveries. Conceptually, these analyses are relevant for any systems biologist or bioinformatician who is interested in integrating large population datasets. Currently, the resource is already of use for scientists studying the HMDP or using GTEx data, and we hope to see updates in the coming years that incorporate more populations and more datatypes, which could make it a general tool for a wide community.

---

## [Referee Report · Reviewer #1 (Public Review)]

Zhou et al. have slightly expanded and improved their web tool from the previous submission, fixing some small issues and adding in additional sets of data from HMDP mice. Essentially, the authors have created a tool that facilitates the integrated analysis of omics datasets (particularly transcriptomics, but could be easily adapted to include proteomics) across tissues.

The strength is that this is new; as far as I know, any other multi-tissue analysis software is relatively ad hoc and it is not easily supported by e.g. SRA/GEO, but rather you'd need to download the multiple datasets and DIY. The authors have now shown some statistically significant (albeit expected from literature) results created using their pipeline. Whether the method will be generally useful for the community depends on its further development and support, but of course whether a project is supported also depends on whether its first publication is accepted - somewhat of a Catch-22 for a reviewer. Right now, the results shown are a convincing proof-of-concept that would likely be of utility mostly to the hosting laboratory and their direct collaborators, but which, with continued development at a similar level of effort, could be more generally useful for the growing number of groups interested in cross-tissue analysis.

---

## [Referee Report · Reviewer #2 (Public Review)]

Summary:

Zhou et al. have revised their previous manuscript, which has greatly improved the quality of the work. Zhou et al. use publicly available GTEx data of 18 metabolic tissues from 310 individuals to explore gene expression correlation patterns within-tissue and across-tissues. Furthermore, they have added an analysis of data from a diverse panel of inbred mouse strains, which allows them to also incorporate data on physiological phenotypes relevant to metabolic signaling between tissues. They now focus on validating their approach to exploring signal in gene co-expression rather than emphasizing unvalidated discoveries. They provide a webtool (GD-CAT) to allow users to explore these data. Focusing more on known biology does result in the study making stronger conclusions from its data. The webtool is also improved, expanded with the mouse data, and of value to the scientific community. Their revision has also corrected key misconceptions from the initial submission and provides greater clarification of the methodologies used.

Strengths:

GTEx as well as the hybrid diversity mouse panel are powerful resource for many areas of biomedicine, and this study represents a valid use of gene co-expression network methodology. They have greatly improved its description and contextualization within the gene co-expression studies. The authors previously did a good job of providing examples confirming known signaling biology and have further improved these. They have largely removed the sections on discovery of novel biology, which is potentially for the better given a lack of follow-up validation, which could be beyond the scope of this manuscript anyway. The webtool, GD-CAT, is easy to use and allows researchers with genes and tissues of interest to perform the same analyses in the GTEx and HMDP data.

Weaknesses:

With the previous version, the primary weaknesses for me were key misconceptions and lack of detail in the methods, which have all been greatly improved. The manuscript could be considered more of a "Resource" than "Research", though there is value in showing how the known biology is reflected in the correlation data and could presumably be paired with validation to discover new biology. Finally, there are sentences here and there that could be rephrased to improve clarity, but overall it is greatly improved.

---

## [Author Response]

The following is the authors’ response to the original reviews.

We thank the reviewers and editors for their time and careful consideration of this study. Nearly every comment proved to be highly constructive and thoughtful, and as a result, the manuscript has undergone major revisions including the title, all figures, associated conclusions and web app. We feel that the revised resource provides a more systematic and comprehensive approach to correlating inter-individual transcript patterns across tissues for analysis of organ cross-talk. Moreover, the manuscript has been restructured to highlight utility of the web tool for queries of genes and pathways, as opposed to focused discrete examples of cherry-picked mechanisms. A few key revisions include:

• Manuscript: All figures have been revised to place to explore broad pathway representation. These analyses have replaced the previous circadian and muscle-hippocampal figures to emphasize ability to recapitulate known physiology and remove the discovery portion which has not been validate experimentally.

• Manuscript: The term “genetic correlation” or “genetically-derived” has been replaced throughout with “transcriptional”, “inter-individual”, or mostly just “correlations”.

• Manuscript: A new figure (revised fig 2) has been added to evaluate the innate correlation structure of data used for common metabolic pathways, in addition an exploration of which tissues generally show more co-correlation and centrality among correlations.

• Manuscript: A new figure (revised fig 4) has been added to highlight the utility of exploring gene ~ trait correlations in mouse populations, where controlled diets can be compared directly. These highlight sex hormone receptor correlations with the large amount of available clinical traits, which differ entirely depending on the tissue of expression and/or diet in mouse populations.

• Web tool: Addition of a mouse section to query expression correlations among diverse inbred strains and associated traits from chow or HFHS diet within the hybrid mouse diversity panel.

• Web tool: Overrepresentation analysis for pathway enrichments have been replaced with score-based gene set enrichment analyses and including network topology views for GSEA outputs.

• Web tool: Associated github repository containing scripts for apps now include a detailed walk-through of the interface and definitions for each query and term.

**Public Reviews:**

**Reviewer #1 (Public Review):**
Zhou et al. have set up a study to examine how metabolism is regulated across the organism by taking a combined approach looking at gene expression in multiple tissues, as well as analysis of the blood. Specifically, they have created a tool for easily analyzing data from GTEx across 18 tissues in 310 people. In principle, this approach should be expandable to any dataset where multiple tissues of data were collected from the same individuals. While not necessary, it would also raise my interest to see the "Mouse(coming soon)" selection functional, given that the authors have good access to multi-tissue transcriptomics done in similarly large mouse cohorts.SummaryThe authors have assembled a web tool that helps analyze multiple tissues' datasets together, with the aim of identifying how metabolic pathways and gene regulation are connected across tissues. This makes sense conceptually and the web tool is easy to use and runs reasonably quickly, considering the size of the data. I like the tool and I think the approach is necessary and surprisingly under-served; there is a lot of focus on multi-omics recently, but much less on doing a good job of integrating multi-tissue datasets even within a single omics layer.What I am less convinced about is the "Research Article" aspect of this paper. Studying circadian rhythm in GTEx data seems risky to me, given the huge range in circadian clock in the sample collection. I also wonder (although this is not even remotely in my expertise) whether the circadian rhythm also gets rather desynchronized in people dying of natural causes - although I suppose this could be said for any gene expression pathway. Similarly for looking at secreted proteins in Figure 4 looking at muscle-hippocampus transcript levels for ADAMTS17 doesn't make sense to me - of all tissue pairs to make a vignette about to demonstrate the method, this is not an intuitive choice to me. The "within muscle" results look fine but panels C-E-G look like noise to me...especially panel C and G are almost certainly noise, since those are pathways with gene counts of 2 and 1 respectively.I think this is an important effort and a good basis but a significant revision is necessary. This can devote more time and space to explaining the methodology and for ensuring that the results shown are actually significant. This could be done by checking a mix of negative controls (e.g. by shuffling gene labels and data) and a more comprehensive look at "positive" genes, so that it can be clearly shown that the genes shown in Fig 1 and 2 are not cherry-picked. For Figure 3, I suspect you would get almost an identical figure if instead of showing pan-tissue circadian clock correlations, you instead selected the electron transport chain, or the ribosome, or any other pathway that has genes that are expressed across all tissues. You show that colon and heart have relatively high connectivity to other tissues, but this may be common to other pathways as well.

Response: We are thankful to the reviewer in their detailed assessment of the manuscript. The comments raised in both the public and suggested reviews clearly improved the revised study and helped to identify limitations. In general, we have removed data suggesting “discovery” using these generalized analyses, such as removing figures evaluating circadian rhythm genes and muscle-hippocampus correlations. These have been replaced with more thorough investigations of tissue correlation structure and potentially identified regions of data sparsity which are important for users to consider. Also, we have added a similar full detailed pipeline of mouse (HMDP) data and highlighted in the manuscript by showing transcript ~ trait correlations of sex hormone receptor genes which differ between organs and diets. Further responses to individual points are also provided below.

**Reviewer #2 (Public Review):**
Summary:Zhou et al. use publicly available GTEx data of 18 metabolic tissues from 310 individuals to explore gene expression correlation patterns within-tissue and across-tissues. They detect signatures of known metabolic signaling biology, such as ADIPOQ's role in fatty acid metabolism in adipose tissue. They also emphasize that their approach can help generate new hypotheses, such as the colon playing an important role in circadian clock maintenance. To aid researchers in querying their own genes of interest in metabolic tissues, they have developed an easy-to-use webtool (GD-CAT).This study makes reasonable conclusions from its data, and the webtool would be useful to researchers focused on metabolic signaling. However, some misconceptions need to be corrected, as well as greater clarification of the methodology used.Strengths:GTEx is a very powerful resource for many areas of biomedicine, and this study represents a valid use of gene co-expression network methodology. The authors do a good job of providing examples confirming known signaling biology as well as the potential to discover promising signatures of novel biology for follow-up and future studies. The webtool, GD-CAT, is easy to use and allows researchers with genes and tissues of interest to perform the same analyses in the same GTEx data.Weaknesses:A key weakness of the paper is that this study does not involve genetic correlations, which is used in the title and throughout the manuscript, but rather gene co-expression networks. The authors do mention the classic limitation that correlation does not imply causation, but this caveat is even more important given that these are not genetic correlations. Given that the goal of their study aligns closely with multi-tissue WGCNA, which is not a new idea (e.g., Talukdar et al. 2016; https://doi.org/10.1016/j.cels.2016.02.002), it is surprising that the authors only use WGCNA for its robust correlation estimation (bicor), but not its latent factor/module estimation, which could potentially capture cross-tissue signaling patterns. It is possible that the biological signals of interest would be drowned out by all the other variation in the data but given that this is a conventional step in WGCNA, it is a weakness that the authors do not use it or discuss it.

Response: Thank you for the helpful and detailed suggestions regarding the study. The review raised some important points regarding methodological interpretations (ex. bicor-exclusive application as opposed to module-based approaches), as well as clarification of “genetic” inferences throughout the study. The comparison to module-based approaches has also now been discussed directly, pointing our considerations and advantages to each. We hope that the reviewer with our corrections to the misconceptions posed, many of which we feel were due to our insufficient description of methodological details and underlying interpretations. The revised manuscript, web portal and associated github provide much more detail and many more responses to specific points are provided below.

**Reviewer #3 (Public Review):**
Summary: A useful and potentially powerful analysis of gene expression correlations across major organ and tissue systems that exploits a subset of 310 humans from the GTEx collection (subjects for whom there are uniformly processed postmortem RNA-seq data for 18 tissues or organs). The analysis is complemented by a Shiny R application web service.The need for more multisystems analysis of transcript correlation is very well motivated by the authors. Their work should be contrasted with more simple comparisons of correlation structure within different organs and tissues, rather than actual correlations across organs and tissues.Strengths and Weaknesses: The strengths and limitations of this work trace back to the nature of the GTEx data set itself. The authors refer to the correlations of transcripts as "gene" and "genetic" correlations throughout. In fact, they name their web service "Genetically-Derived Correlations Across Tissues". But all GTEx subjects had strong exposure to unique environments and all correlations will be driven by developmental and environmental factors, age, sex differences, and shared and unshared pre- and postmortem technical artifacts. In fact we know that the heritability of transcript levels is generally low, often well under 25%, even studies of animals with tight environmental control.This criticism does not comment materially detract for the importance and utility of the correlations-whether genetic, GXE, or purely environmental-but it does mean that the authors should ideally restructure and reword text so as to NOT claim so much for "genetics". It may be possible to incorporate estimates of chip heritability of transcripts into this work if the genetic component of correlations is regarded as critical (all GTEx cases have genotypes).Appraisal of Work on the Field: There are two parts to this paper: 1. "case studies" of cross-tissue/organ correlations and 2. the creation of an R/Shiny application to make this type of analysis much more practical for any biologist. Both parts of the work are of high potential value, but neither is fully developed. My own opinion is that the R/Shiny component is the more important immediate contribution and that the "case studies" could be placed in the context of a more complete primer. Or Alternatively, the case studies could be their own independent contributions with more validation.

Response: We thank the reviewer for their supportive and helpful comments. The discussion of usage of the term “genetic” has been removed entirely from the manuscript as this point was made by all reviewers. Further, we have revised the previous study to focus on more detailed investigations of why transcript isoforms seemed correlated between tissues and areas where datasets are insufficient to provide sufficient information (ex. Kidney in GTEx). As the reviewer points out, the previous “case studies” were unvalidated and incomplete and as a result, have been replaced. Additional points below have been revised to present a more comprehensive analyses of transcript correlations across tissues and improved web tool.

**(Recommendations For The Authors):**
As this manuscript is focused on the analytical process rather than the biological findings, the reviewer concerns are not a fundamental issue to subsequent acceptance of the paper, but some of the examples will need to be replaced or double-checked to ensure their biological and statistical relevance. To raise the scope and interest of the method developed, it would be seen very positively to include additional datasets, as the authors seem to have intended to have done, with a non-functional (and highlighted as such) selection for mouse data. Establishing that the authors can easily - and will easily - add additional datasets into their tool would greatly raise the reviewers' confidence in the methodology/resource aspect of this paper. This may also help address the significant concerns that all three reviewers raised with the biological examples, e.g. that GTEx data is so uncontrolled that studying environmentally-influenced traits such as circadian rhythm may be challenging or even impossible to do properly. Adding in a more highly controlled set of cross-tissue mouse data may be able to address both these concerns at once, i.e. the resource concern (can the website easily be updated with new data) and the biological concern (are the results from these vignettes actually statistically significant).
**Reviewer #1 (Recommendations For The Authors):**
Comments, in approximately reverse order of importance1. Some figure panels are not referenced in the text, e.g. Fig 1B and Figure 2E.Response: Thank you for pointing this out. We have revised every figure in the manuscript and additionally gone through to make sure every panel is referenced in the text.1. The authors mention "genetic data" several times but I don't see anything about DNA. By "genetic data" do you mean "transcriptome expression data," or something else?

Response: This is an important point, also raised by all 3 reviewers. We have clarified in the abstract, results and discussion that correlations are between transcripts. As a result, all mentions of “genetics” or “genetic data” has been removed, with the exception of introducing mouse genetic reference panels.

1. For Figure 3, the authors look at circadian clock data, but the GTEx data is from all sorts of different times of day from across the patient cohort depending on when the donor died, and I don't see this metadata actually mentioned anywhere. I see Arntl Clock and all the other circadian genes are highly coexpressed in each tissue (except not so strong in liver) but correlation across tissue seems more random. Also hypothalamus seems to be very strongly negatively correlated with spleen, but this large green block doesn't have significance? That is surprising to me, since the sample sizes are all equivalent I would expect any correlation remotely close to -1.0 to be highly significant.

Response: The reviewer raises several important points with regard to the source of data and underlying interpretations. We have added a revised Fig 2, suggesting that representation of gene expression between tissues can be strongly biased by nature of samples (ex. differences in data that is available for each tissue) and also discussed considerations of the nature of sample origin in the limitations section. We have also used some of these points when introducing rationale for using mouse population data. As a result of comments from this reviewer and others, we have removed the circadian rhythm analysis and muscle-hippocampal figures from the revised study; however, specifically mentioned these cohort differences in the discussion section (lines 294-298). Circadian rhythm terms are also evaluated in Fig 2 and consistent with the reviewers concerns, less overall correlations are observed between transcripts across tissues when compared to other common GO terms assessed.

1. Figure 4, this is all transcript-level data, so it is confusing to see protein nomenclature used, e.g. "expression of muscle ADAMTS17" should be "expression of muscle ADAMTS17" (ADAMTS17 the transcript should be in italics, in case the formatting is removed by the eLife portal). Same for FNDC5. In the figures you do have those in italics, so it is just an issue in the manuscript text. In general please look through the text and make sure whether you are referring really to a "gene," "transcript," or "protein." For instance, Figure 1 legend I think should be "A, All transcripts across the ... with local subcutaneous and muscle transcript expression." I know people still sometimes use "gene expression" to refer to transcripts, but now that proteomics is pretty mainstream, I would push for more careful vocabulary here.

Response: Thank you for pointing these out. While we have replaced Fig 4 entirely as to limit the unvalidated discovery or research aspects of the paper, we have gone through the text and figures to check that the correct formatting is used for references to human genes (capitalized italics) or the newly-included mouse genes (lower-case italics).

1. "Briefly, these data were filtered to retain genes which were detected across individuals where individuals were required to show counts > 0 in 1.2e6 gene-tissue combinations across all data." I don't quite understand the filtering metric here - what is 1.2 million gene-tissue combinations referring to? 20k genes times 18 tissues times 310 people is ~100 million measurements, but for a given gene across 310 people * 18 tissues that is only ~6000 quantifications per gene.

Response: We apologize for this oversight, as the numbers were derived from the whole GTEx dataset in total and not the tissues used for the current study. We have clarified this point in the revised manuscript (methods section in Datasets used) and also removed confusing references to specific numbers of transcripts and tissues unless made clear.

1. Generally I think your approach makes sense conceptually but... for the specific example used in e.g. figure 4, this only makes sense to me if applied to proteins and not to transcripts. Looking at the transcript levels per tissue for genes which are secreted could be interesting but this specific example is confusing, as is the tissue selected. I would not really expect much crosstalk between the hippocampus and the muscle, especially not in terms of secreted proteins.

Response: This is a valid point, also raised by other reviewers. While we wanted to highlight the one potentially-new (ADAMTS7) and two established proteins (FNDC5 and ERFE) and their correlations, the fact that this direct circuit remains to be validated led us to replace the figure entirely. The point raised about inference of protein secretion compared to action; however, has been expanded upon in the results and discussion. We now show that complexities arise when using this approach to infer mechanisms of proteins which are primarily regulated post-transcriptionally. We provide a revised Supplemental Fig 4 showing that this general framework, when applied to expression of INS (insulin), almost exclusively captured pathways leading to its secretion and not action.

1. It's not clear to me how correction for multiple testing is working in the analyses used in this manuscript. You mention q-values so I am sure it was done, I just don't see the precise method mentioned in the Methods section.

Response: We apologize for this oversight and have included a specific mention of qvalue adjustment using BH methods, where our reasoning was the efficiency in run-time (compared to other qvalue methods). In addition, we provide a revised Fig 2 which suggests that innate correlation structure exists between tissues for a variety of pathways which should be considered. We also compare several empirical bicor pvalues and qvalue adjustments directly between these large pathways where much of the innate tissue correlation structure does appear present when BH qvalue adjustments are applied (revised Fig 2A).

1. The piecharts in Figure 1 are interesting - I would actually be curious which tissues generally have closer coexpression. This would be an absolutely massive number of pairwise correlations to test, but maybe there is a smarter way to do it? For instance, for ADIPOQ, skeletal muscle has the best typical correlation, but would that be generally true just that many adipose genes have closer relationship between the two tissues?

Response: This comment inspired us to perform a more systematic query of global gene-gene correlation structures, which is now shown as the revised Fig 2A. With respect to ADIPOQ, the reviewer is correct in that there does appear to be a general pattern of muscle genes showing stronger correlation with adipose genes. We emphasize and discuss there in the revised manuscript to point out that global trends of tissue correlation structure should be taken into account when looking at specific genes. Much of this innate co-correlation structure could be normalized by the BH qvalue adjustment (above); however, strongly correlated pathways like mitochondria showed selective patterns throughout thresholds (revised Fig 2A). Further, we analyze KEGG terms and general correlation structures (revised Fig 2B) to point out the converse, that some tissues are just poorly represented. Interpretation of correlated genes from these organ and pathway combinations should be especially considered in the framework that their poor representation in the dataset clearly impacted the global correlation structures. We have added these points to both results and discussion. In sum, we feel that this was a critical point to explore and attempted to provide a framework to identify/consider in the revised manuscript.

1. The pathway enrichments in Figure 1 are more difficult for me to interpret, e.g. for ADIPOQ, the scWAT pathways make sense, but the enriched skeletal muscle pathways are less clearly relevant (rRNA processing?? Not impossible but no clear relevance either). What are the significances for these pathway enrichments? Is it even possible to select a gene that has no peripheral pathway enrichment, e.g. if you take some random Gm#### or olfactory receptor gene and run the analysis, are you also going to see significant pathways selected, as pathway enrichment often has a trend to overfit? The "within organ" does seem to make sense, but I am also just looking at 4 anecdotes here and it is unclear whether they are cherry picked because they did make sense. That is, it's unclear why you selected ADIPOQ and not APOE or HMGCR or etc. I also don't figure out how I can make these pathway enrichment plots using your website. I do get the pie chart but when I try the enrichment analysis block (NB: typo on your website, it says "Enrich-E-ment Analysis" with an extra E) I always get that "the selected tissue do not contain enough genes to generate positive the enrichment." (Also two typos in that phrase; authors should check and review extensively for improvements to the use of English.) After trying several genes I eventually got it to work. I think there is some significant overfitting here, as I am pretty sure that XIST expression in the white adipose tissue has nothing to do with olfactory signalling pathways, which are the top positive network (but with an n = 4 genes).

Response: Several good points within this comment. (1) the pathway enrichments have been revised completely. The reviewer provided a helpful suggestion of a rank-based approach to query pathways, as opposed to the previous over-representation tests. After evaluating several different pathway enrichment tools based on correlated tissue expression transcripts, a rank- and weight-based test (GSEA) captured the most physiologic pathways observed from known actions of select secreted proteins. Therefore, revised pathway enrichments and web-tool queries unitize a GSEA approach which accounts for the rank and weight determined by correlation coefficient. In implementing these new pathway approaches, we feel that pathway terms perform significantly better at capturing mechanisms. (2) With respect to the selection genes, we wanted to provide a framework for investigating genes which encode secreted proteins that signal as a result of the abundance of the protein alone. This is a group-bias; however, and not necessarily reflective of trying to tackle the most important physiologic mechanisms underlying human disease. We agree with the reviewer in those evaluating genes such as APOE and cholesterol synthesis enzymes present an exciting opportunity, our expertise in interpretation and mechanistic confirmation is limited. (3) We have gone through the revised manuscript and attempted to correct all grammatical and/or spelling mistakes.

1. The network figures I get on your website look actually more interesting than the ones you have in Figure 2, which only stay within a tissue. Making networks within a tissue is pretty easy I think for any biologist today, but the cross-tissue analysis is still fairly hard due to the size of the datasets and correlation matrices.

Response: We greatly appreciate the reviewer’s enthusiasm for the network model generation aspect. We have tried to improve the figure generation and expanded the gene size selection for network generation in the web tool, both within and across tissues. We are working toward allowing users to select specific pathway terms and/or tissue genes to include in these networks as well, but will need more time to implement.

1. I get a bug with making networks for certain genes, e.g. XIST - Liver does not work for plotting network graphs. Maybe XIST is a suppressed gene because it has zero expression in males? It is an interesting gene to look at as a "positive control" for many analyses, since it shows that sample sexing is done correctly for all samples.

Response: The reviewer recognized a key consideration in underlying data structure for GTEx. In the revised manuscript, we evaluated tissue representation (or lack thereof) being a crucial factor in driving where significant relationships cannot be observed in tissues such as kidney, liver and spleen (Fig 2). Moreover, the representation of females (self-reported) in GTEx is less-than half of males (100 compared to 210 individuals). We have emphasized this point in the discussion where we specifically pointed out the lack of XIST Liver correlation being a product of data structure/availability and not reflecting real biologic mechanisms. We expanded on this point by highlighting the clear sex-bias in terms of representation.

1. On the network diagram on your website, there doesn't seem to be any way to zoom in on the website itself? You can make a PDF which is nice but the text is often very small and hard to read.

Response: We have revised the web interface plot parameters to create a more uniform graph.

1. On a related note, is it possible to output the raw data and gene lists for the network graph? I would want to know what are those genes and their correlation coefficient.

Response: We have enabled explore as .pdf or .svg graphics for the network and all plots. In addition, following pie chart generation at the top of the web app, users now have the ability to download a .csv file containing the bicor coefficients, regression pvalues and adjusted qvalues for all other gene-tissue combinations.

1. Some functionality issues, e.g. on the "Scatter plot" block, I input a gene name again here. Shouldn't this use the same gene selected already at the top of the page? It seems confusing to again select the gene and tissue here, but maybe there is a reason for that.

Response: It would be more intuitive to only display genes from a given selected tissue for scatterplots; however, we chose to keep all possible combinations with the [perhaps unnecessary] option of reselecting a tissue to allow users to query any specific gene without having to wait to run the pathways for all that correspond to a given tissues.

1. Figure 4H should also probably be Figure 1A.

Response: Good point, the revised Fig 1A is now a summary of the web tool

I realize I have written a fairly critical review that will require most of the figures to be redone, but I think the underlying method is sound and the implementation by and end-user is quite simple, so I think your group should have no trouble addressing these points.

Response: Your comments were really helpful and we feel that the tool has significantly improved as a result. So, we are thankful to the time and effort put toward helping here.

**Reviewer #2 (Recommendations For The Authors)**
Comments on the use of "genetic correlation"• The use of "genetic correlation" in title and throughout the manuscript is misleading. Should broadly be replaced with "gene expression correlation". Within genetics, "genetic correlation" generally refers to the correlation between traits due to genetic variation, as would be expected under pleiotropy (genetic variation that affects multiple traits). Here, I think the authors are somewhat conflating "genetic" (normally referring to genetic variation) with "gene" (because the data are gene expression phenotypes). I don't think they perform any genetic analysis in the manuscript. I hope I don't sound too harsh. I think the paper still has merit and value, but it is important to correct the terminology.

Response: This was an important clarification raised by all reviewers. We apologize for the oversight. As a result, all mentions of “genetics” or “genetic data” has been removed, with the exception of introducing mouse genetic reference panels. These have generally been replaced with “transcript correlations”, “correlations” or “correlations across individuals” to avoid confusion.

• The authors note an important limitation in the Discussion that correlations don't imply a specific causal model between two genes, and furthermore note that statistical procedures (mediation and Mendelian randomization) are dependent on assumptions and really only a well-designed experiment can completely determine the relationship. This is a very important point that I greatly appreciate. I think they could even further expand this discussion. The potential relationships between gene A and gene B are more complex than causal and reactive. For example, a genetic variant or environmental exposure could regulate a gene that then has a cascade of effects on other genes, including A and B. They belong to a shared causal pathway (and are potentially biologically interesting), but it's good to emphasize that correlations can reflect many underlying causal relationships, some more or less interesting biologically.

Response: We thank the reviewer for pointing this out. We have expanded both the results and discussion sections to mention specifically how correlation between two genes can be due to a variety of parameters, often and not just encompassing their relationship. We mention the importance of considering genetic and environmental variables in these relationships as well which we feel will be an important “take-home message” for the reader. These points were also explored in the revised Fig 2 in terms of investigating broad pathway gene-gene correlation structures. As noted by the reviewer, contexts such as circadian rhythm or other variables in the data which are not fixed show much less overall significance in terms of broad relationships across organs.

• It would be good for the authors to provide more context for the methods they use, even when they are fully published. For example, stating that biweight midcorrelation (bicor) is an approach for comparing to variables that is more robust to outliers than traditional correlations and is commonly used with gene co-expression correlation.

Response: Thank you for pointing this out. A lack of method description was also an important reason for lack of clarity on other aspects so we have done our best to detail what exact approaches are being implemented and why. In the revised manuscript, we mention the usage if bicor values to limit influence of outlier individuals in driving regressions, but also point out that it is still a generalized linear model to assess relationships. We hope that the revised methods and expanded git repositories which detail each analysis provide much more transparency on what is being implemented.

• Performing a similar analysis based on genetic correlation is an interesting idea, as it would potentially simplify the underlying causal models (removing variation that doesn't stem from genetic variants). I don't expect the authors to do this for this paper because it would be a significant amount of work (fitting and testing genetic correlations are not as straightforward). But still, an interesting idea to think about, and individuals in GTEx are genotyped I believe. Could be mentioned in the Discussion.

Response: Absolutely. While we did not implement and models of genetic correlation (despite misusing the term) in this analysis. We have added to the discussion on how when genetic data is available, these approaches offer another way to tease out potentially causal interactions among the large amount of correlated data occurring for a variety of reasons.

Comments on use of the term "local" and "regression"• "Local" is largely used to mean within-tissue, so how correlated gene X in tissue Y is with other genes in tissue Y. I think this needs to be defined explicitly early in the manuscript or possibly replaced with something like "within-tissue".

Response: We have replaced al “local” mentions with “within-tissue” or simply name the tissue that the gene is expressed to avoid confusion with other terms of local (ex a transcript in proximity to where it is encoded on the genome).

• "Regression" is also used frequently throughout, often when I think "correlation" would be more accurate. It's true that the regression coefficient is a function of the correlation between X and Y, but I don't think actual regression (the procedure) applies here. The coefficients being used are bicor, which I don't think relates as cleanly to linear regression.

Response: Thank you for pointing this out. A lack of method description was also an important reason for lack of clarity on other aspects so we have done our best to detail what exact approaches are being implemented and why. In the revised manuscript, we mention the usage if bicor values to limit influence of outlier individuals in driving correlations, but also point out that it is still a generalized linear model to assess relationships. Further, we have removed usage of “regression” when referencing bicor values. We hope that the revised methods and expanded git repositories which detail each analysis provide much more transparency on what is being implemented.

• "Further, pan-tissue correlations tend to be dominated by local regressions where a given gene is expressed. This is due to the fact that within-tissue correlations could capture both the regulatory and putative consequences of gene regulation, and distinguishing between the two presents a significant challenge" (lines 219-223). This sentence includes both "local" and "regressions" (and would be improved by my suggested changes I think), but I also don't fully understand the argument of "regulatory and putative consequences". I think the authors should elaborate further. In the examples, the within-tissue correlations do look stronger, suggesting within-tissue regulation that is quite strong and potentially secondary inter-tissue regulation. If that's the idea, I think it can be stated more clearly.

Response: Thank you for pointing this out. We have revised the sentence to state the following:

Further, many correlations tend to be dominated by genes expressed within the same organ. This could be due to the fact that, within-tissue correlations could capture both the pathways regulating expression of a gene, as well as potential consequences of changes in expression/function, and distinguishing between the two presents a significant challenge. For example, a GD-CAT query of insulin (INS) expression in pancreas shows exclusive enrichments in pancreas and corresponding pathway terms reflect regulatory mechanisms such as secretion and ion transport (Supplemental Fig 4).

We feel that this point might not be intuitive, so have included a new figure (Supplemental Fig 4) which contains the tissue correlations and pathways for INS expression in pancreas. These analyses show an example where co-correlation structure seems almost entirely dominated by genes within the same organ (pancreas) and GSEA enrichments highlight many known pathways which are involved in regulating the expression/secretion of the gene/protein. We hope that this makes the point more clearly to the reader.

Additional comments on Results:• I would break the titled Results sections into multiple paragraphs. For example, the first section (lines 84-129) has a few natural breakpoints that I noticed that would potentially make it feel less over-whelming to the reader.

Response: We have broken up the results section into separate paragraphs in the revised manuscript. In addition, we have gone through to try and make sure that the amount of information per block/sentence focuses on key points.

• "Expression of a gene and its corresponding protein can show substantial discordances depending on the dataset used" (line 224 of Results). This is a good point, and the authors could include citations here of studies that show discordance between transcripts and proteins, of which there are a good number. They could also add some biological context, such as saying differences could reflect post-translational regulation, etc.

Response: Thank you for the supportive comment. We have referenced several comprehensive reviews of the topic, each of which contain tables summarizing details of mRNA-protein correlation. The revised discussion sentence is as follows:

Expression of a gene and its corresponding protein can show substantial discordances depending on the dataset used. These have been discussed in detail39–41, but ranges of co-correlation can vary widely depending on the datasets used and approaches taken. We note that for genes encoding proteins where actions from acute secretion grossly outweigh patterns of gene expression, such as insulin, caution should be taken when interpreting results. As the depth and availability of tissue-specific proteomic levels across diverse individuals continues to increase, an exciting opportunity is presented to explore the applicability of these analyses and identify areas when gene expression is not a sufficient measure.

1. Liu, Y., Beyer, A. & Aebersold, R. On the Dependency of Cellular Protein Levels on mRNA Abundance. Cell 165, 535–550 (2016).

2. Maier, T., Güell, M. & Serrano, L. Correlation of mRNA and protein in complex biological samples. FEBS Letters 583, 3966–3973 (2009).

3. Buccitelli, C. & Selbach, M. mRNAs, proteins and the emerging principles of gene expression control. Nat Rev Genet 21, 630–644 (2020).

• In many ways, this work has similar goals to many studies that have performed multi-tissue WGCNA (e.g., Talukdar et al. 2016; https://doi.org/10.1016/j.cels.2016.02.002). In this manuscript, WGCNA's conventional approach to estimating robust correlations (bicor) is used, but they do not use WGCNA's data reduction/clustering functionality to estimate modules. Perhaps the modules would miss the signaling relationships of interest, being sort of lost in the presence of stronger signals that aren't relevant to the biological questions here. But I think it would be good for the authors to explain why they didn't use the full WGCNA approach.

Response: This is an important point and we also feel that the previous lack of methodological details and discussion did a poor job at distinguishing why module-based approaches were not used. We wanted to be careful not to emphasize one approach being superior/inferior to another, rather point out the different considerations and when a direct correlation might inform a given question. As the reviewer points out, our general feeling is that adopting a simple gene-focused correlation approach allows users to view mechanisms through the lens of a single gene; however, this is limited in that these could be influenced by cumulative patterns of correlation structure (for example mitochondria in revised Fig 2A) which would be much more apparent in a module-based approach. This comment, in combination with the other listed above, was our motivation in exploring cumulative patterns of gene-gene correlations in the revised Fig 2. In the revised manuscript, we expanded on the results and discussion section to highlight utility of these types of approaches compared to module-based methods:

The queries provided in GD-CAT use fairly simple linear models to infer organ-organ signaling; however, more sophisticated methods can also be applied in an informative fashion. For example, Koplev et al generated co-expression modules from 9 tissues in the STARNET dataset, where construction of a massive Bayesian network uncovered interactions between correlated modules6. These approaches expanded on analysis of STAGE data to construct network models using WGCNA across tissues and relating these resulting eigenvectors to outcomes42. The generalized approach of constructing cross-tissue gene regulatory modules presents appeal in that genes are able to be viewed in the context of a network with respect to all other gene-tissue combinations. In searching through these types of expanded networks, individuals can identify where the most compelling global relationships occur. One challenge with this type of approach; however, is that coregulated pathways and module members are highly subjective to parameters used to construct GRNs (for example reassignment threshold in WGCNA) and can be difficult in arriving at a “ground truth” for parameter selection. We note that the WGCNA package is also implemented in these analyses, but solely to perform gene-focused correlations using biweight midcorrelation to limit outlier inflation. While the midweight bicorrelation approach to calculate correlations could also be replaced with more sophisticated models, one consideration would be a concern of overfitting models and thus, biasing outcomes.

Additional comments on Discussion:• In the second paragraph of the Discussion (lines 231-244), the authors mention that GD-CAT uses linear models to compare data between organs and point to other methods that use more complex or elaborate models. It's good to cite these methods, but I think they could more directly state that there are limitations to high complexity models, such as over-fitting.

Response: Thank you for this suggestion. We have added a line (above) mentioning the overfitting concern.

Comments on Methods:• The described gene filtration in the Methods of including genes with non-zero expression for 1.2e6 gene-tissue combinations is confusing. If there are 310 individuals and 18 tissues, for a given gene, aren't there only 5,580 possible data points? Might be helpful to contextualize the cut-off in terms of like the average number of individuals with non-zero expression within a tissue.

Response: We apologize for this error. This number was pasted from a previous dataset used and not appropriate for this manuscript. In general, we have removed specific mentions of total number of gene_tissue correlation combinations, as these numbers reflect large but almost meaningless quantifications. Instead, we expanded the methods in terms of how individuals and genes filtered.

• More details should be given about the gene ontology/pathway enrichment analysis. I suspect that a set-based approach (e.g., hypergeometric test) was used, rather than a score-based approach. The authors don't state what universe of genes were used, i.e., the overall set of genes that the reduced set of interest is compared to. Seems like this could or should vary with the tissues that are being compared. A score-based approach could be interesting to consider (https://www.biorxiv.org/content/10.1101/060012v3), using the genetic correlations as the score, as this would remove the unappealing feature of sets being dependent on correlation thresholds. This isn't something that I would demand of the published paper, but it could be an appealing approach for the authors to consider and confirm similar results to the set-based analysis.

Response: This is an important point. Following this suggestion, we evaluated several different rank- and weight-based pathway enrichment tools, including FGSEA and others. Ultimately, we concluded that GSEA performed significantly better at (1) recapitulating known biology of select secreted protein genes and (2) leveraging the large numbers of genes occurring at qvalue cutoffs without having to further refine (ex. in the previous overrepresentation tests). For this reason, all pathway enrichments in the web tools and manuscripts not contain GSEA outputs and corresponding pathway enrichments or network graph visualizations. Thank you for this suggestion.

Comments on figures:• I think there is a bit of a missed opportunity to use the figures to introduce and build up the story for readers. For example, in Figure 1, plotting ADIPOQ expression against a correlated gene in adipose (local) as well as peripheral tissues. This doesn't need to be done for every example, but I think it would help readers understand what the data are, and what's being detected before jumping into higher level summaries.

Response: Thank you, this point also builds on others which recommended to restructure the manuscript and figures. In the revised manuscript, we first introduce the web tool (which was last previously), and immediately highlight comparisons of within- and across-organ correlations, such as ADIPOQ. We feel that the revised manuscript presents a superior structure in terms of demonstrating the key points and utility of looking at gene-gene correlations across tissues.

• Figures 1 and 4 are missing the color scale legend for the bar plots, so it's impossible to tell how significant the enrichments are.

Response: We apologize for the oversight. The pathways in the revised Fig 1 detail pathway network graphs among the top pathways which should make interpretation more intuitive. We have also gone through and made sure that GSEA enrichment pvalues are now present for all figures including pathways (revised Fig 1, Fig 3 and supplemental Fig 4).

• The Figure 2 caption says that edges are colored based on correlation sign? Are there any negative correlations (red)? They all look blue to me. The caption could also state that edge weight reflects correlation magnitude (I assume). It would be ideal to include a legend that links a range of the depicted edge weights to their genetic correlation, though I don't know how feasible that may be depending on the package being used to plot the networks.

Response: Good catch. We included in the revised manuscript the network edge parameters:Network edges represent positive (blue) and negative (red) correlations and the thicknesses are determined by coefficients. They are set for a range of bicor=0.6 (minimum to include) to bicor=0.99

Related to seeing a dominant pattern of positive correlations, we agree that this observation is fascinating and gene-gene correlations being dominated by positive coefficients will be the topic of a closely-following manuscript from the lab

• Figure 4A would be more informative as boxplots, which could still include Ssec score. This would allow the reader to get a sense of the variation in correlation p-value across all hippocampus transcripts.

Response: Related to comments from this reviewer and others, we have removed the previous Fig 4 entirely from the manuscript to emphasize the ability of these gene-gene correlations to capture known biology and limit the extend of unvalidated “suggested” new mechanisms.

Comments on GD-CAT• The online webtool worked nicely for me. It was easy to use and produce figures like in the manuscript. One suggestion is show data points in the scatter plot rather than just the regression line (if that's possible currently, I didn't figure it out). A regression line isn't that interesting to look at, but seeing how noisy the data look around it is something humans can usually interpret intuitively.

Response: Thank you so much. We are excited that the web tool works sufficiently. We have also revised the individual gene-gene correlation tab to show individual data points instead of simple regression lines.

Minor comments:

Response: Thank you for these detailed improvements

• This sentence is awkwardly constructed: "Here, we surveyed gene-gene genetic correlation structure for ~6.1x10^12 gene pairs across 18 metabolic tissues in 310 individuals where variation of genes such as FGF21, ADIPOQ, GCG and IL6 showed enrichments which recapitulate experimental observations" (lines 68-70). It's an important sentence because it's where in the Abstract/Introduction the authors succinctly state what they did, thus I would re-work it to something like: "Here, we surveyed gene expression correlation structure..., identifying genes, such as FGF21, ADIPOQ, GCG and IL6, that possess correlation networks that recapitulate known biological pathways."

Response: The numbers of pairs examined and dataset size have been removed for clarity and we have revised this statement and results as a whole

• Prefer swapping "signal" for "signaling" in line 53 of Abstract/Introduction.

Response: Done

• Remove extra period in line 208 of Results.

Response: Removed

• Change "well-establish" to "well-established" in line 247 of Discussion.

Response: Replaced

• Missing commas in line 302 of Methods.

Response: added

• Missing comma in line 485 of Figure 3 caption.

Response: The previous Fig 3 has been removed

• Typo in title of Figure 3E (change "Perihperal" to "Peripheral")

Response: Thank you, changed

• Add y-axis label to y-axis labels (relative cell proportions) to Supplemental Figures 1-3.

Response: These labels have been added

**Reviewer #3 (Recommendations For The Authors):**
Minor technical comment: The authors refer to correlations between genes when they actually mean correlations between GTEX transcript isoform models. It is exceedingly important to keep this distinction clear in the reader's mind, a fact that is emphasized by the authors themselves when they comment on the potential value of similar proteomic assays to evaluate multiorgan system communication. GTEx has tried to do proteomics but I do not know of any open data yet.

Response: Thank you for this point. We have gone through the manuscript and replaced “gene correlations” with “transcript” or other similar mentions. Related to the comment on GTEx proteomics, this is an important point as well. As the reviewer mentions, proteomics has been performed on GTEx data; however, given that this dataset contains only 6 sparsely-represented individuals, analyses such as the ones highlighted in our study remain highly limited. We have added the following to the discussion:As the depth and availability of tissue-specific proteomic levels across diverse individuals continues to increase, an exciting opportunity is presented to explore the applicability of these analyses and identify areas when gene expression is not a sufficient measure. For example, mass-spec proteomics was recently performed on GTEx42; however, given that these data represent 6 individuals, analyses utilizing well-powered inter-individual correlations such as ours which contain 310 individuals remain limited n applications.

The R/Shiny companion application: The community utility of this application would be greatly improved by a link to a primer and more basic functionality. The Github site is a "work in progress" and does not include a readme file or explanation (that I could find) on the license.

Response: Thank you, we are excited that the apps operate sufficiently. We have revised the github repository entirely to contain a full walk-through of app details and parameter selections. These are meant to walk users through each step of the pipeline and discuss what is being done at each step. We agree that this updated github repository allows users to understand the details of the R/Shiny app in much more detail. We also made all the app scripts, datasets, markdown/walkthrough files and docker image fully available to enhance accessibility.